 

# Cerebellar patients have intact feedback control that can be leveraged to improve reaching

Amanda M Zimmet[1,2], Di Cao[3], Amy J Bastian[1,4†], Noah J Cowan[3†*]

[1]Kennedy Krieger Institute, Baltimore, United States; [2]Department of Biomedical Engineering, Johns Hopkins University, Baltimore, United States; [3]Department of Mechanical Engineering and Laboratory for Computational Sensing and Robotics, Johns Hopkins University, Baltimore, United States; [4]Department of Neuroscience, Johns Hopkins University, Baltimore, United States

**Abstract** It is thought that the brain does not simply react to sensory feedback, but rather uses an internal model of the body to predict the consequences of motor commands before sensory feedback arrives. Time-delayed sensory feedback can then be used to correct for the unexpected—perturbations, motor noise, or a moving target. The cerebellum has been implicated in this predictive control process. Here, we show that the feedback gain in patients with cerebellar ataxia matches that of healthy subjects, but that patients exhibit substantially more phase lag. This difference is captured by a computational model incorporating a Smith predictor in healthy subjects that is missing in patients, supporting the predictive role of the cerebellum in feedback control. Lastly, we improve cerebellar patients' movement control by altering (phase advancing) the visual feedback they receive from their own self movement in a simplified virtual reality setup.

**\*For correspondence:**
ncowan@jhu.edu

[†]These authors contributed equally to this work

## Introduction

Humans normally rely on a balance of feedback and predictive control mechanisms to make smooth and accurate movements. Proprioceptive and visual feedback are necessary for determining body postures at the beginning and end of a movement and can be used to guide slow movements accurately. However, feedback is time delayed, and thus it never represents the current state of the body during movement. Because of this, it is thought that we depend on internal models of the body that are built based on prior experience. These models can be rapidly accessed and thus provide a fast internal prediction system to estimate how a movement will unfold, enabling us to better understand where our limbs are at any given moment. This allows us to make fast and accurate movements despite long-latency feedback.

People with cerebellar damage show a characteristic pattern of incoordination during movement that is referred to as ataxia. When reaching, they make curved movements that miss intended targets and require multiple corrections. This pattern of over- and undershooting a target (dysmetria) and oscillatory corrections (intention tremor) are hallmarks of cerebellar ataxia. One hypothesis that might explain ataxia is that the predictive estimation and control provided by cerebellar circuits is dysfunctional or lost (*Miall et al., 2007*; *Wolpert et al., 1998*).

Normally, the estimation of limb state (e.g. position and velocity) benefits from integrating proprioceptive measurements with an internal predictive control model during a movement (*Adamovich et al., 1998*; *Fuentes and Bastian, 2010*; *Paillard and Brouchon, 1974*). However, patients with cerebellar damage do not seem to receive this benefit (*Bhanpuri et al., 2013*; *Weeks et al., 2017*). Worse, it is possible that their predictive model actually conveys incorrect state information during active movements, which could corrupt rather than enhance proprioceptive

estimation of limb state. This difficulty of predicting the future state of limbs during active movement leads to movements that are poorly directed and scaled, requiring ongoing corrections to reach a goal location.

Patients with cerebellar ataxia may rely more heavily on visual feedback to correct dysmetric movements (*Beppu et al., 1987*; *Beppu et al., 1984*; *Day et al., 1998*). A drawback of visual feedback is that it is slower than proprioception; intention tremor in these patients' movements is thought to stem from dependence on time-delayed visual feedback to make corrections (*Day et al., 1998*). However, it is not known how well they incorporate this visual feedback into their movements. Is visual feedback control impaired? Or does the dysmetria stem solely from errors in predictive (i.e. feedforward) control (*Bhanpuri et al., 2014*; *Manto et al., 1994*; *Smith and Shadmehr, 2005*)? A gait study by Morton and Bastian suggested that cerebellar patients could use feedback information to make reactive corrections during split-belt treadmill walking in order to maintain stability, but lacked the ability to adapt their predictive motor patterns (*Morton and Bastian, 2006*). Other studies have indirectly supported the notion that patients can incorporate some level of visual feedback control in reaching (*Day et al., 1998*; *Smith et al., 2000*).

In this study, we used behavioral experiments and computational modeling to test for any impairment in visual feedback control (correction based on measured error) and disambiguate it from previously described impairments in feedforward control (*Bares et al., 2007*; *Broersen et al., 2016*). Subjects performed a visuomotor task which required them to track an unpredictable target (*Roth et al., 2011*). This method allowed us to determine that cerebellar patients can integrate visual feedback control similarly to healthy, age-matched control subjects. We then hypothesized that we could exploit intact feedback control to reduce dysmetria. Specifically, we provided an acceleration-dependent alteration to visual feedback of a hand cursor. This alteration serves to compensate for the phase lags introduced by time delay, allowing patients to use visual feedback more effectively. This real-time controller successfully reduced patients' dysmetria.

## Materials and methods

### Experimental model and subject details

A total of 17 patients with cerebellar deficits and 14 age-matched controls were tested in one or more of the following experiments. Patients were excluded if they had any clinical or MRI evidence of damage to extra-cerebellar brain structures, or clinical evidence of dementia, aphasia, peripheral vestibular loss, or sensory neuropathy. The age-matched controls were clinically screened for any neurological impairments. Experiments 1–2 tested 11 cerebellar patients (seven male, four female) and 11 age-matched controls (four male, seven female). Patient and control ages were within +/- 3 years of age. The one exception is a pairing of a 78-year-old patient with a 71-year-old control. Experiment 3 tested 12 cerebellar patients (eight male, four female) and 12 age-matched controls (three male, nine female), and all patient and control ages were within +/- 3 years of age. Subjects gave informed consent according to the Declaration of Helsinki. The experimental protocols were approved by the Institutional Review Board at Johns Hopkins University School of Medicine. Each subject used his or her dominant arm for all tasks. The only exception was one unilaterally affected cerebellar patient who was instructed to use her affected, non-dominant, hand. We quantified each subject's ataxia using the International Cooperative Ataxia Rating Scale (ICARS) (*Trouillas et al., 1997*). Because this study involves upper limb reaching behavior, we calculated an Upper Limb ICARS sub-score comprising the sum of the upper-limb kinetic function elements of the test. The results of the ICARS as well as demographic information for all patients are shown in *Table 1*.

### Experimental apparatus, experiments, and task instructions

For all experiments, subjects were seated in a KINARM exoskeleton robot (BKIN Technologies Ltd., Kingston, Ontario, Canada), shown in *Figure 1A*, which provided gravitational arm support while allowing movement in the horizontal plane. A black video screen occluded the subjects' view of their arm movements. The shoulder position was fixed at a 75° angle, as shown in *Figure 1C*. The wrist joint was also fixed; therefore, the elbow joint was the only freely mobile joint. Data were recorded at 1 kHz. The KINARM system exhibits a cursor delay. Using a high-speed camera, we measured this delay (0.0458 s) and took the delay into account in modeling the human control system.

**Table 1.** The cerebellar patient population that was tested.

Not all subjects completed all experiments. Some subjects completed the experiments over two visits. Upper Limb ICARS contains the sum of the upper-limb kinetic function elements of the test (ICARS subsections 10–14, out of 20).

| Subject no. | Patient age | Sex | Pathology | Icars | Upper limb ICARS | Experiments |
|---|---|---|---|---|---|---|
| P1 | 44 | M | SCA8 | 62 | 17 | 1,2,3 |
| P2 | 52 | M | ADCAIII | 28 | 8 | 3 |
| P3 | 53 | M | Sporadic | 59 | 19 | 1,2,3 |
| P4 | 55 | F | SCA8 | 41 | 17 | 3 |
| P5 | 55 | M | OPCA | 46 | 14 | 1,2 |
| P6 | 60 | M | MSA-C | 63 | 14 | 1,2,3 |
| P7 | 62 | F | Sporadic | 36 | 16 | 1,2 |
| P8 | 62 | M | SCA6/8 | 62 | 19 | 1,2,3 |
| P9 | 63 | M | SCA6 | 41 | 10 | 3 |
| P10 | 64 | F | SCA6 | 58 | 13 | 1,2 |
| P11 | 64 | M | ADCAIII | 11 | 3 | 1,2,3 |
| P12 | 65 | M | Idiopathic | 34 | 10 | 3 |
| P13 | 66 | F | SCA6 | 41 | 11 | 1,2 |
| P14 | 67 | F | Left cerebellar stroke | 27 | 10 | 1,2,3 |
| P15 | 69 | F | ADCAIII | 52 | 13 | 3 |
| P16 | 72 | F | SCA6 | 49 | 18 | 3 |
| P17 | 78 | M | Sporadic (Vermis Degen.) | 39 | 6 | 1,2 |

A 1 cm diameter white dot (cursor) was projected onto the display to indicate the subject's index fingertip position. The display was calibrated so that the projected dot position aligned with the subject's fingertip position. Subjects performed three experiments:

- Experiment 1: sum-of-sines tracking task.
- Experiment 2: single-sines tracking task.
- Experiment 3: discrete reaches with acceleration-dependent feedback.

For Experiments 1 and 2 (tracking tasks), subjects were instructed to try to keep the cursor in the center of the target, a 1.5 cm diameter green dot (*Figure 1C*). The target angle followed a pseudo-random sum-of-sines (*Figure 1D*) or single-sine pattern. Each trial was 100 s long. We tested three different Feedback Gain conditions in the sum-of-sines (Experiment 1) task: 1.35 (cursor moves 35% farther than hand), 0.65 (cursor motion is attenuated by 35%), and 1.0 (veridical feedback; *Figure 1B*). Subjects were not informed that there was a gain change.

For Experiment 3 (discrete reaches), the robot moved the arm to align the cursor with the red dot start position (55° elbow angle) before each reach. After approximately 2 s, the red dot disappeared, and a green target dot (1.5 cm diameter, at 85° elbow angle) appeared elsewhere on the screen. Subjects were instructed to bring the white cursor dot to the center of the green target dot (a 30° flexion movement) as smoothly and accurately as possible without over or undershooting the target. If the fingertip did not reach within 1.75 cm of the target within 800 ms, the green dot changed to blue and subjects were instructed to complete the reach (trials were not repeated); the subjects were also asked to try to move faster on the next trial. The next trial started after the subject kept the fingertip in the target dot for two continuous seconds.

Healthy control subjects performed five practice trials and patients performed 10 practice trials to familiarize themselves with the procedure. If subjects were still confused about the procedure after the practice trials, they were given more practice trials until they were comfortable with the procedure.

Some subjects completed only a subset of the experiments due to time constraints (see *Table 1*). For every experiment, between trials, subjects were told to relax while the robot moved the subject's hand to bring the cursor to the start position.

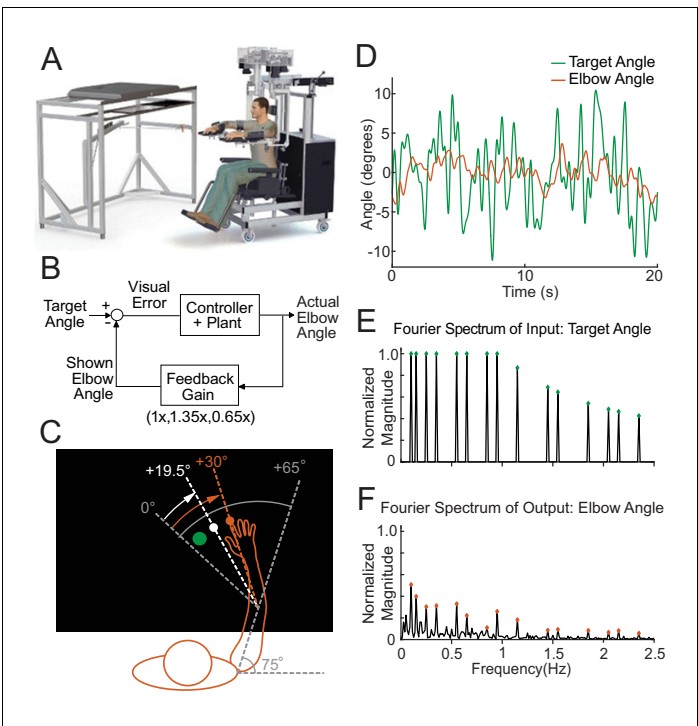

**Figure 1.** The experimental paradigm. (A) The Kinarm Exoskeleton robot system (BKIN Technologies Ltd.) where all tasks are executed. Image is taken from *Tyryshkin et al., 2014*. (B) This block diagram shows our experimental paradigm for experiment 1. The actual elbow angle (orange in C) is multiplied by a feedback gain to compute the shown elbow angle (white dot in C). (C) A schematic view of the Kinarm screen in the 0.65 Feedback Gain condition. The subject's shoulder is locked at a 75° angle. The green dot is the target dot. The green dot oscillates according to the sum-of-sines pattern along an arc with a radius equivalent to the forearm + hand + finger length. The sum-of-sines pattern is centered about the 0° angle. This 0° centerline is at a 65° angle from the subject's straight arm position. The subject could not see their fingertip (orange dot) or arm position. The angle of the white dot (visible to the subject) is equal to 0.65 times the actual fingertip angle in orange (0.65 · 30°=19.5°). (D) Sample data (20 s) for a single cerebellar patient showing the angles traversed by the target dot during the sum-of-sines task (green) and by the fingertip (orange, averaged over five trials). (E) The normalized Fourier spectrum of the input sum-of-sines signal (green trace in D). (F) The spectrum of the response by a single cerebellar patient over five trials (orange trace in D).

## Experimental design and statistical analysis

### Generating target trajectories (Experiments 1 and 2)

The goal of Experiment 1 was to determine how well patients and controls could track an unpredictable sum-of-sines stimulus and probe their ability to use visual feedback for movement control. The target trajectory comprised 15 sine waves whose frequencies were prime multiples of 0.05 Hz, namely 0.1, 0.15, 0.25, 0.35, 0.55, 0.65, 0.85, 0.95, 1.15, 1.45, 1.55, 1.85, 2.05, 2.15, and 2.35 Hz. Phases of each sinusoid were randomized. Each trial was 100 s long, comprising five replicates of the sum-of-sines trajectory. The sinusoids were scaled so that the *angular velocity* of each sinusoid never exceeded 720°/s, while the *positional* amplitude of each sine wave never exceeded 2°. The non-harmonic relation of the component sinusoids created a temporally complex target motion with 45 degrees of freedom (amplitude, phase, and frequency for 15 sinusoids) that repeats every 20 s; a portion of the trajectory is shown in *Figure 1D*, wherein it can be observed that the signal trajectory includes multiple seemingly random turn-arounds within even very short timescales. This complexity, coupled with the long period, makes this target motion far less predictable than a simple single-sine motion (*Miall and Jackson, 2006*; *Poulton, 1974*; *Roth et al., 2011*). Lastly, the dramatic increase in tracking phase lag presented in this paper (see Results) for sum-of-sine responses recapitulates the main finding of *Roth et al., 2011*, confirming that from a practical point of view subjects were less able to predict the movement trajectories. Nevertheless, this signal design allowed good signal-

to-noise ratio at each frequency while avoiding excessively large or fast motions of the target, making it well suited as a probe signal for tracking behavior (*Roth et al., 2011*). The normalized magnitude of the Fourier spectrum of our input signal is shown in the top half of in *Figure 1E*. Five trials were conducted for each condition tested.

In Experiment 2, we wanted to determine how well patients and controls could track a predictable single sine wave. Every other frequency tested in the sum-of-sines condition was tested as a single, standalone sine wave: 0.1, 0.25, 0.55, 0.85, 1.15, 1.55, 2.05, and 2.35 Hz. The amplitude of each sine wave matched (component-wise) the sum-of-sines experiments. Subjects were once again instructed to keep the cursor in the target dot as much as possible. The order of presentation of the stimuli was randomized for each subject.

## Estimating frequency responses (Experiments 1 and 2)

All analyses were performed using custom scripts (*Zimmet et al., 2020*) in MATLAB (The Mathworks Inc, Natick, MA, USA). To obtain the steady-state frequency response of the subject, the first period (20 s) was discarded as transient, leaving four periods per trial. The data were visually inspected for unusual activities that were not representative of the subject's typical behavior. For example, if the subject turned away from the screen to cough or talk, movement would cease for a few seconds and the data from those few seconds were removed from further analysis. The movement trajectories for a given subject were then averaged across trials at each time instant, excluding any deleted data. The averaged data were linearly detrended and converted to the frequency domain via the discrete Fourier transform (DFT). An example of the result of this process for a cerebellar patient is shown in *Figure 1F* (the frequency domain data is complex valued at each frequency so only the magnitude is shown).

For Experiment 1 (sum of sines), to estimate a subject's frequency response, we calculated the ratio of the DFT of the subject's movement to the DFT of the underlying target trajectory at each of the 15 frequencies in the sum-of-sines. The result of this calculation is 15 complex numbers, called *phasors*, representing the frequency response estimate for a given subject. The magnitude of each phasor is the *gain* and the angle is the *phase* of the subject's response at that frequency.

The processing of the data for Experiment 2 (single sines) was identical to the processing of the sum-of-sines data, except that there was no averaging across trials and no manual inspection/filtering of the data for unusual activities. This is because we were only looking at one frequency from each sine wave trial, and any error caused by an unusual, aperiodic activity (such as pausing) would introduce very little power at any individual frequency.

## Phasor plot (Experiments 1 and 2)

To visualize the frequency response, we used Phasor Plots as shown in *Figure 2A*. In these plots, the gain is the radial distance from the origin, and phase lag is the clockwise angle of the data with respect to the horizontal axis. Perfect tracking at a given frequency corresponds to unity gain and zero phase lag, as illustrated by the large dot in *Figure 2B*. The distance between any point on the phasor plot and this dot provides a measure of the amplitude of the (sinusoidal) error signal at that particular frequency as a proportion of the input signal amplitude.

Additionally, this visualization can be enhanced with a circle with a 0.5 radius centered at +0.5 along the horizontal axis as shown in *Figure 2B*. Points on this circle correspond to the ideal response gain for minimizing error given a particular phase lag (*Roth et al., 2011*). Points on this circle represent the gain values that minimize error assuming the subject cannot further minimize phase lag. Points outside the circle indicate the subject is moving too much for a given phase lag; reducing the amplitude of movement would achieve a lower tracking error and would do so with less effort. Points within the circle may represent striking a balance between effort and error in that the subject is not moving enough to fully minimize their error given a certain phase lag. Lastly, points that lie on the unit circle, traced in bold black in *Figure 2C*, would indicate that the subject is replicating the sinusoidal component exactly, albeit at a phase shift.

## Phasor plot scaling (Experiment 1)

If a subject were to 'fully' respond to an applied visual gain, the subject would need to scale his or her movement so that the visual output appeared the same as it did in the veridical feedback

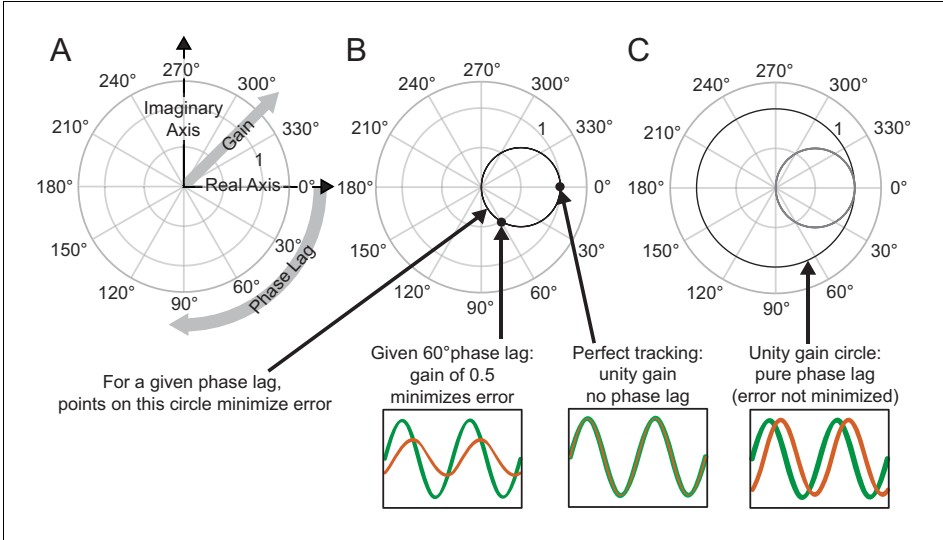

**Figure 2.** Interpreting phasor plots. (A–C) Brief explanation of phasor plots, which are explained in depth in Methods in 'Phasor Plot.' (A) Illustration of the relationship between the real and imaginary components of the complex numbers and the gain and phase lag. (B) The bold black circle illustrates the gains that minimize error given a particular phase lag (veridical feedback condition only). (C) Points on the bold black circle illustrate a different control strategy than what was asked of our subjects. Points on this circle replicate the input signal without minimizing error between the input and output signals (veridical feedback condition only) due to the phase lag.

condition, although there is no constraint (or instruction) to do so. In reality, a subject may scale his or her movements more than this, less than this, or even not at all. To quantify this, we fit a Scaling Factor that, when multiplied by the frequency response phasors for a given Feedback Gain condition, best aligned the phasors with those of the veridical condition. We only included the five lowest frequencies in this calculation because at these frequencies subjects generally appeared to have robust responses (based on visual inspection of the data). At high frequencies, however, response magnitudes were low and therefore fitting the Scaling Factor could be dominated by noise.

Scaling Factor (dependent measure) was analyzed using a two-way ANOVA, testing for the effects of two independent variables, group (patient vs. control) and Feedback Gain (0.65 and 1.35). If a main effect was found to be significant, we calculated an effect measure ($\eta_p^2$) using the Measures of Effect Size toolbox for Matlab (*Hentschke and Stüttgen, 2011*).

## Modeling (using data from Experiment 1)

To capture the visuomotor tracking response of both cerebellar patients and age-matched healthy controls, we adapted the classic McRuer gain-crossover model: a delayed, scaled integrator which specifically assumes that the subject is responding to an unpredictable stimulus (*McRuer and Krendel, 1974*). The McRuer gain-crossover model is highly simplified, and in its original incarnation was meant to capture the sensorimotor frequency response near the gain crossover frequency (i.e. where the open-loop gain of the combined plant and controller has unit magnitude). The idea is that near this frequency, closed-loop robustness is governed by the amount of phase lag; if that phase lag exceeds 180°, the closed-loop system becomes unstable. In a neighborhood of the gain-crossover frequency, *McRuer and Krendel, 1974* recognized that subjects tended to have an open-loop gain whose magnitude dropped off inversely with frequency (like an integrator, or k/s), and the phase lag was a bit more than 90° (the integrator adds 90 degrees, and a delay in their model increases the phase lag as a function of frequency). We adopt a version of this simplified model to facilitate the interpretation of data (Figure 4).

We hypothesized that the patient and control models would be equivalent except for the magnitude of the feedback delay (Figure 4A). To test this, we considered a more general class of models that allows for distinct parameters for the patients and healthy control subjects, and includes the

possibility of distinct delays on the visual measurement of target motion from that of self-movement feedback (which can incorporate for example, proprioception, which generally incurs a lower latency than visual). To determine the simplest combined model of patients and controls which provided both a good and consistent fit without overfitting, we used a model selection process similar to that described in *Madhav et al., 2013* based on Experiment 1 data (sum-of-sines). In the most flexible model, all patient and control parameters were allowed to vary independently with eight free parameters: $k_{patient}$, $k_{control}$, Visual Gain$_{patient}$, Visual Gain$_{control}$, Visual Delay$_{patient}$, Visual Delay$_{control}$, Feedback Delay$_{patient}$, and Feedback Delay$_{control}$ (*Figure 4—figure supplement 1*). The following model selection procedure aimed to determine which of those eight parameters should be free and which should be yoked together to most parsimoniously fit the data. All combinations of yoked parameters were tested, subject to yoking gains with gains, and delays with delays. We also tested a few degenerate model structures (such as a pure gain or delay, with no feedback); some of these model structures allowed for the elimination of one or more of the blocks within the model structure.

The following procedure was repeated for each possible model (note: here a 'model' includes both patient and control submodels). For each subject, we represented their frequency response as an array of 15 complex numbers (one for each of the frequencies tested in the sum-of-sines experiment). We began by pulling 10 of the 11 age-matched patient–control pairs from the full dataset. Then, we took the average of those 10 patients' (and, separately, controls) frequency responses at each tested frequency. Using the MATLAB function 'fminsearch,' we used these average frequency response functions to fit the current model parameters to the average data by minimizing the frequency-domain (FD) error:

$$\text{FD-error} = \left\| \text{control}_{\text{avg}} - \text{control}_{\text{modelfit}} \right\|^2 + \left\| \text{patient}_{\text{avg}} - \text{patient}_{\text{modelfit}} \right\|^2 \tag{1}$$

where *control* and *patient* in the above equation were arrays with 15 complex values, corresponding respectively to frequency responses of healthy control subjects and patients at the 15 frequencies tested. This was repeated 100 times with different initial parameter values selected for the 'fminsearch' function, increasing the likelihood of finding a global minimum. The model parameters that generated the lowest overall error were kept (as well as the magnitude of the error). This was repeated for each patient–control pair, yielding 11 sets of model parameters and 11 leave one out error (loo-error) values, defined as:

$$\text{loo-error} = \left\| \text{control}_{\text{leftout}} - \text{control}_{\text{modelfit}} \right\|^2 + \left\| \text{patient}_{\text{leftout}} - \text{patient}_{\text{modelfit}} \right\|^2 \tag{2}$$

The leave one out error values were averaged together to create the overall leave one out error for that model structure. This was our gauge for the ability of the model to capture the responses. Using the 11 sets of the model parameters for that particular model (i.e. one set of parameters for each subject pair that was left out), we created an 11 × 8 matrix, each row of which corresponding to the parameters for a given leave-one-out fit. The elements of the matrix were the residuals between each of the eight parameters and their average value. We calculated the maximum singular value of this matrix as a gauge for model consistency. This was repeated for all potential models. Finally, by modifying our fitting procedure above, we used bootstrapping to generate confidence intervals for the parameter values in our best model. Specifically, we randomly sampled eleven times with replacement from the eleven subjects. This was repeated 1000 times to generate 1000 sets of parameter values. From these values, we found the 95% confidence intervals.

'Essentially all models are wrong, but some are useful' (*Gep and Draper, 1987*) so rather than choosing a single 'correct' model, we scrutinized all models that exhibit a good tradeoff of model fit and model consistency. We also considered the number of free parameters (parsimony). Lastly, we examined whether a given model produced physiologically realistic parameters; models with non-physiological parameters would suggest inadequate model structure leading to parameter bias. We determined the common features and subtle differences between these good models and the outcome of these meta analyses are described in Results.

We converted each frequency-domain model (which includes patient and control submodels) to state space in Matlab, and simulated the model pair using the lsim command (*Figure 4—figure supplement 1D*). The input was the sum-of-sines target angle trajectory used in experiments. The model output produced a distinct prediction for the elbow-angle trajectory for each group (patients and controls; Figure 4C). These simulated elbow-angle responses were compared to subject responses

as follows. We averaged across trials for each subject to compute a single time-domain response for each patient and each age-matched control; we then averaged across subjects in each group to compute a mean elbow-angle response for patients and a mean elbow-angle response for controls. Given the simulated model responses and actual patient and control mean responses, the time-domain (TD) error was calculated as follows:

$$\text{TD-error} = \sqrt{\frac{1}{T} \int_0^T \left(\text{control(t)}_{\text{mean}} - \text{control(t)}_{\text{model}}\right)^2 dt} + \sqrt{\frac{1}{T} \int_0^T \left(\text{patient(t)}_{\text{mean}} - \text{patient(t)}_{\text{model}}\right)^2 dt} \quad (3)$$

where the integration is computed as a Riemann sum over the last $T = 80$ seconds of the 100s trial period, with a sampling time of 1ms. The TD-error was normalized so that the maximum error was 1 for visualization purposes.

## Comparing phase lags across cohorts and conditions (Experiments 1 and 2)

We tested whether the cerebellar group showed a different pattern of phase lag across frequencies compared to controls. Specifically, we compared the phase lag for the lowest common frequencies (0.10, 0.25, 0.55, 0.85, 1.15) in the sum-of-sines and single-sine conditions across the two groups. We hypothesized that the control group would be able to use prediction and follow the single sine wave with little phase lag compared to the cerebellar patients. We expected that there would be less of a difference between groups when they followed the sum-of-sines. We used a parametric two-way ANOVA for circular data called the Harrison-Kanji test for this analysis (*Berens, 2009*).

## Acceleration-Dependent Feedback (Experiment 3)

To examine the effect of modified visual cursor feedback in VR, we implemented Acceleration-Dependent Feedback, where the cursor angle was set to follow the elbow angle plus an acceleration-dependent term:

$$\text{CursorAngle(t)} = \text{ElbowAngle(t)} + k_a \frac{d^2}{dt^2} \text{ElbowAngle(t)} \quad (4)$$

To implement this Acceleration-Dependent Feedback, we used the KINARM's real-time computer to calculate the average of the previous 100 elbow angular acceleration values as an approximation of the instantaneous acceleration (at 1 kHz sampling, this resulted in a ~ 50 ms delay). This acceleration estimate was multiplied by the Acceleration-Dependent Feedback Gain ($k_a$), added to the real elbow angle, and the position of the cursor was displayed on the Kinarm screen at this new, slightly shifted angle. In practical terms, this moving-average-filtered acceleration-dependent feedback amounts to a high-pass filter, providing anticipatory (i.e. phase-leading) feedback to the user when $k_a > 0$, and providing phase-lagging feedback when $k_a < 0$.

To find a patient-specific feedback gain $k_a$, cerebellar patients performed blocks of 10 reaches where this gain was held constant. Each patient started with $k_a = 0$ (veridical feedback) for their first block of reaches. Individual reaches were categorized as hypometric, on-target, or hypermetric. Based on the mode of this categorization for the 10 reaches, the gain was increased, remained the same, or decreased, as determined by the Parameter Estimation by Sequential Testing (PEST) algorithm (see Implementation of the PEST Algorithm (Experiment 3), below) (*Taylor and Creelman, 1967*).

In addition, control and patient participants performed blocks of five reaches at nine specific gains in this order: $k_a = 0$ (veridical), 0.005, 0.010, 0.015, 0.020, −0.005, −0.010, −0.015, and −0.020. Participants were given five trials of one Feedback Gain before being exposed to the next Feedback Gain on the list. Note that only 6 of the 12 patients completed this experiment due to time constraints.

## Quantifying dysmetria for discrete reaches (Experiment 3)

We quantified dysmetria in this single-joint task by measuring the elbow angle at which the subject made his or her first correction. We determined the angle of the first correction by finding when the velocity crossed zero after the initiation of the reach. The angle of the elbow at this time point is the angle of the first correction. This angle was divided by the goal angle of 30°. In a smooth and

accurate reach, no correction would be needed and the 'angle of first correction' would be the target angle; in this scenario, the ratio between the 'angle of first correction' and the goal angle would be 1. If the result was greater than 1.03, the reach was categorized as hypermetric, and likewise, if the result was less than 0.97, the reach would be categorized as hypometric. Results between 0.97 and 1.03 were classified as 'on target' reaches.

## Implementation of the PEST algorithm (Experiment 3)

The Parameter Estimation by Sequential Testing (PEST) algorithm was used to iteratively determine the best Acceleration-Dependent Feedback Gain values to apply in Experiment 3 by analyzing the history of the responses to different applied values (*Taylor and Creelman, 1967*). If the mode of the initial set of reaches was hypermetric, the Acceleration-Dependent Feedback Gain would be decreased. Likewise, it would be increased if the mode of the set was hypometric. The initial step size used for the PEST algorithm was 0.01. The maximum step size for the PEST algorithm was limited to 0.01 and the minimum step size was 0.0035. The maximum Acceleration-Dependent Feedback Gain was set to +/- 0.02. The PEST algorithm was terminated when two blocks of the same Acceleration-Dependent Feedback Gain yielded a mode of reaches classified as 'on target.' Alternatively, if this portion of the experiment took longer than approximately 20 min or the subject was experiencing fatigue, the most successful Acceleration-Dependent Feedback Gain was selected based on those which had already been tested.

## Results

### Experiment 1: cerebellar patients and age-matched controls respond similarly to rescaling of visual self-motion feedback

To probe participants' ability to use visual feedback, we challenged cerebellar patients and age-matched controls to keep a cursor within a target dot that was following an unpredictable trajectory. The normalized Fourier spectrum (magnitude only) of the target trajectory is shown in *Figure 1E* and the spectrum of a single subject's response is shown in *Figure 1F*. Note that this subject had clear peaks at all of the frequencies of target movement in the veridical condition (1x).

Subjects also were exposed to two Feedback Gain conditions where the cursor was presented at 0.65x or 1.35x. A phasor plot for each Feedback Gain condition is shown for a single cerebellar patient in *Figure 3A*. Points with the smallest phase lag (~30°) are responses from the lowest frequencies (0.1 Hz). Responses at the five lowest frequencies are marked with solid dots. As a function of increasing input frequency, the phase lag increases and the plotted points move progressively clockwise around the origin. Note in *Figure 3A* that the example subject scales his movement, especially at the lowest frequencies, in response to the Feedback Gain applied. For the 0.65 Feedback Gain condition, he increases the scaling of his movement in comparison to the veridical condition, as expected. Similarly, in the 1.35 Feedback Gain condition, he decreases the scaling of his movement in comparison to the veridical condition.

For each patient, we computed a Scaling Factor that best scaled responses at the lowest five frequencies (Materials and methods); to visualize how well this Scaling Factor represents our data, the phasor data from an example cerebellar patient in the 1.35 Feedback Gain condition shown in *Figure 3A* is multiplied by its scaling factor to yield the corresponding line in *Figure 3B*. If the scaling factor is a good fit for the data, the data should rest on top of the veridical line. Note that the phase is not modified by this scaling computation, so we should not expect to see an improvement in phase alignment in this second plot.

*Figure 3C* shows the Scaling Factors for all subjects for each Feedback Gain condition. We conducted a one-sided t-test to determine whether patients scaled their gain in the hypothesized direction against the null hypothesis that patients' scaling factors would be 1 (i.e. they would not respond to the applied feedback gain). Both groups scaled up (patients: p=0.03, t = −2.1, DOF = 10; controls: p=$4\times10^{-6}$, t = −8.3, DOF = 10) or scaled down (patients: p=0.005, t = 3.23, DOF = 10; controls p=0.0002, t = 5.3, DOF = 10) their movement, commensurate with the respective decrease or increase of the cursor Feedback Gain. We performed a two-way ANOVA, confirming that the effect of Feedback Gain (0.65 vs. 1.35) on Scaling Factor was significant ($F(1,40)$=59.7, p=$2\times10^{-9}$, $\eta_p^2$=0.7), whereas the effect of group (patient vs. control) on Scaling Factor was not significant ($F(1,40)$=0.05,

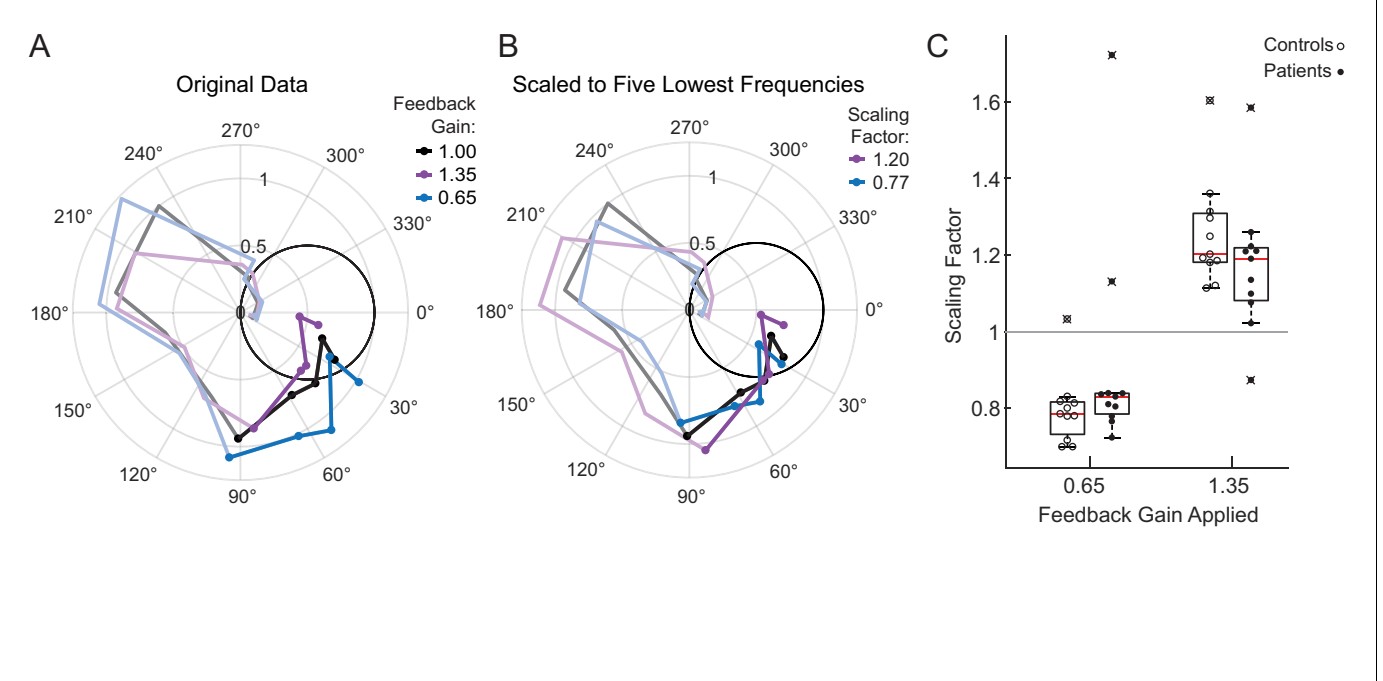

**Figure 3.** Motion rescaling under different feedback gains. (**A**) Sample phasor plot data from a single cerebellar patient under all three feedback gain conditions. The dots represent the frequency response at the lowest five frequencies. The solid line traces through the responses at all tested frequencies. The dimmed lines show the frequency responses at the other frequencies tested beyond the five lowest frequencies. (**B**) Scaled phasor plot data from the same single cerebellar patient. The gain value from D has been multiplied by the scaling factor shown in the legend here to create an overlaid phasor plot. The higher the Scaling Factor, the less effort (in terms of movement magnitude) the subject expended in that condition. (**C**) Patients and controls respond similarly to the applied Feedback Gain, indicating that they incorporate visual feedback similarly. The higher the Scaling Factor, the less effort (in terms of movement magnitude) the subject expended in that condition. Outliers are marked with an x. The outlier subject with the highest Scaling Factor in the 0.65 condition was from the same subject who had the lowest Scaling Factor in the 1.35 condition.

p=0.83), with no significant interaction between group and gain (*F*(1,40)=3.57, p=0.07). We accounted for the samples being dependent on one another because each subject was exposed to both the 0.65 and 1.35 gain conditions. The effect size was computed using the Hentschke and Stüttgen toolbox (*Hentschke and Stüttgen, 2011*).

Note that the subjects typically exhibited a Scaling Factor of around 1.2 in the 1.35 Feedback Gain condition. Intuitively, this indicates that subjects rescaled their movements as if to compensate perfectly for an experimentally applied Feedback Gain of 1.2; had they compensated perfectly, their scaling factor would have been 1.35. Likewise, for a Feedback Gain of 0.65, the Scaling Factor was typically approximately 0.8, again implying that subjects compensated, albeit not fully, for the visual rescaling (*Figure 3*). Critically, the patients' responses to the feedback scaling were comparable to that of age-matched controls, suggesting that they were able to use visual feedback in a substantively similar manner.

## Cerebellar patient performance best captured by long latency closed-loop model

In computational terms, one can interpret the data from Experiment one to mean that cerebellar patients had a functional feedback loop in their control system. Here we asked if they are using the same 'control structure' (i.e. a model of the interplay between sensory feedback, external sensory input, and motor output) as age-matched controls. Because the cerebellar and age-matched control groups scaled their movements similarly in Experiment 1, one might hypothesize that they were using a similar control structure. However, the scaling was measured relative to each subject's baseline movement in the veridical condition and was not a measure of their overall error performance on the task; for example, issues of phase lag would not be captured by the scaling analysis. Thus, here we use our findings from Experiment 1 along with the model selection procedure described by

*Madhav et al., 2013* to determine any differences in control structures used by patients and age-matched controls.

Consider the hypothesized model depicted in *Figure 4A*. In this model, the brain calculates the error between perceived elbow angle and target angle, as shown in the subtraction calculation in the feedback diagram. The model merges the cascade of the subject's internal controller and mechanical plant (arm and robot dynamics); the combined plant and controller is treated as a classic McRuer gain-crossover model—a scaled integrator—a model based on the assumption that the subject is responding to an unpredictable stimulus (*McRuer and Krendel, 1974*). Our hypothesis is that all model parameters would be equivalent between patients and age-matched controls, except that patients would have a feedback delay commensurate with their visual processing time and controls would have a lower latency feedback delay commensurate with their proprioceptive feedback processing time (*Bhanpuri et al., 2013*; *Cameron et al., 2014*; *Crevecoeur and Scott, 2013*; *Izawa and Shadmehr, 2008*). This is because we expect that patients would rely more on (slower) visual feedback to compensate for their deficient estimation of limb state. Visuomotor delay during smooth pursuit tracking is generally much faster than the time required for movement initiation, and estimates for such visuomotor tracking delay vary, but would be expected in the range of 110 to 160 ms (*Brenner and Smeets, 1997*; *Day and Lyon, 2000*; *Franklin and Wolpert, 2008*; *Haith et al., 2016*; *Pruszynski et al., 2016*; *Saunders and Knill, 2003*).

To test this hypothesized model, we performed model fitting and model selection on all possible model configurations (*Figure 4—figure supplement 1A,B*). Our hypothesized model was just one possibility and the parameter values of the models were not constrained a priori. The models were compared based on joint consideration of model consistency and model fit. Model fitting and selection produced five models, called Best 4 (Lowest Variance), Best 4 (Lowest Error), Best 5 (Lowest Variance), Best 5 (Lowest Error), and Best 6, that were nearly equivalent in their trade-off between model inconsistency and model fitting error (*Figure 4—figure supplement 1C*). Model Best 6 added one more free parameter compared to Best 5 (Lowest Variance) and was the only one of the top five models that allowed for variation in the feedback delay for controls; this addition yielded a small delay (~39 ms) and while only providing a small enhancement in model fit-error, shown in both frequency domain plot and time domain validation (*Figure 4—figure supplement 1C* and *Figure 4—figure supplement 1D*). Thus, out of parsimony, we eliminated model Best 6 considered both Best 4 and both Best 5 models as the top models. These top models were extremely similar in their structure and parameters (*Figure 4—figure supplement 1B* and *Table 2*). Indeed, in all four top models, the feedback delay was zero for control subjects, substantially shorter than cerebellar patients. This suggests that controls can rely on an internal model of their hand position and/or proprioceptive feedback to make corrections for their future movements while cerebellar patients must rely on delayed visual feedback (see Discussion).

Rather than picking up a single best model, we drew the following general conclusions from the modeling. The Final Model (*Figure 4B*) was distilled from the top four models to show the general features among them, together with the range of parameters derived from these models. In addition to the parameter values having consistent values across the top four models, they also have intuitively plausible values (*Table 2*). While the true parameters have biological limitations, we did not restrict their values during the fitting procedure. This provides a useful diagnostic tool, since unrealistic parameter fits would indicate inadequate model structure: parameter bias is a hallmark of model deficiency. In the four top models, Visual Delays for controls were found to be 141–147 ms, which is physiologically plausible. Interestingly, this delay was much shorter than for patients, whose Visual Delays were 181–211 ms. Critically, patients exhibited longer response delay both on the visual measurement of target motion and that of self-movement feedback, compared to controls. The top models also suggest that Feedback Delay is shorter than Visual Delay for both patients and controls, although for two of top models, the patients exhibited equivalent Visual and Feedback Delays. Visual Gain values are all approximately 0.40 which also seems reasonable given a visual inspection of the data: subjects do not appear to be attempting to replicate the full magnitude of the signal, but some smaller portion of the signal. We further examined the frequency responses of patients and controls (*Figure 4—figure supplement 2A*). Patients exhibited substantially greater phase lag at high frequencies than controls; at low frequencies, both populations exhibited very little phase lag, but, surprisingly, controls exhibited slightly more phase lag than patients in this frequency

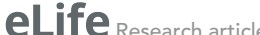

**Figure 4.** Modeling. (**A**) Our modeling framework was based on the McRuer Crossover model (**McRuer and Krendel, 1974**). This simple model structure lumps the controller and plant as a scaled integrator, and assigns different delays to the measurement of the target from that of self-movement feedback. We hypothesized that the patient and control models would be equivalent except for the magnitude of the feedback delay. The Visual Gain is necessary because, given the complexity and speed of the signal, participants were unable to match the full magnitude of the signal. (**B**) After our model selection procedure, the Final Model structure was distilled from the top models to capture the general features shared among them. It is similar to our hypothetical model, but with subtle differences: the feedback delay for the controls was determined to be zero instead of equivalent to the proprioceptive delay, the feedback delay for patients was shorter than or equal to the visual delay, and the visual delay for patients was slightly longer than healthy controls. (**C**) Time domain visualization of subject and model responses for a typical 15 s time window of our experiments. The same visual target trajectory (green) was played to all patients and controls and was used in simulation. Individual time-domain elbow-angle responses (light gray) are shown for the 11 control subjects (top plot) and 11 patients (bottom plot). The average subject responses (orange, solid) match with reasonable accuracy the simulated model responses (orange, dashed). The simulated model structure used here is the modeled named 'Best 4 (Lowest Err)' in **Figure 4—figure supplement 1B**.

The online version of this article includes the following figure supplement(s) for figure 4:

**Figure supplement 1.** Model fitting and model selection.
**Figure supplement 2.** Frequency responses.

**Table 2.** Model parameters with standard deviation for the top models (Best 5 and Best 4).
Variables with the same color within a row are yoked together for that model configuration. Top models are similar; the feedback delay was smaller than or equal to the visual delay for the patients, and zero for controls, and the patient visual delay was longer than that of the healthy controls.

| Model | $k_{patient}$ | $k_{control}$ | Visual Gain$_{patient}$ | Visual Gain$_{control}$ | Visual Delay$_{patient}$ (ms) | Visual Delay$_{control}$ (ms) | Feedback Delay$_{patient}$ (ms) | Feedback Delay$_{control}$ (ms) |
|---|---|---|---|---|---|---|---|---|
| Best 5 (Lowest Err) | 2.8 (0.09) | 2.8 (0.09) | 0.38 (0.01) | 0.41 (0.02) | 211 (0.008) | 143 (0.003) | 143 (0.003) | 0 |
| Best 5 (Lowest Var) | 2.6 (0.08) | 2.6 (0.08) | 0.37 (0.01) | 0.42 (0.02) | 183 (0.005) | 147 (0.003) | 183 (0.005) | 0 |
| Best 4 (Lowest Err) | 2.7 (0.10) | 2.7 (0.10) | 0.39 (0.01) | 0.39 (0.01) | 210 (0.007) | 141 (0.003) | 141 (0.003) | 0 |
| Best 4 (Lowest Var) | 2.5 (0.09) | 2.5 (0.09) | 0.40 (0.01) | 0.40 (0.01) | 181 (0.005) | 145 (0.003) | 181 (0.005) | 0 |

range. Thus, there was a phase 'cross over' frequency between patients and controls, which was also captured by our top models (*Figure 4—figure supplement 2B*).

As a final test of our modeling approach, we examined how our Best 4 (Lowest Err) Model's parameters changed when we applied variations in Feedback Gain as shown in *Table 3*. Recall that in the 1.35 Feedback Gain condition, the dot moves more than the person's hand position; therefore, we expect the subject to move slightly less than they do in the veridical feedback condition because they do not need to move as much to get the same visual output (i.e. we expect k to decrease). Similarly, we expect the subject to attempt to replicate a greater portion of the signal because it is easier to do so. Thus, we expect the Visual Gain (the amount of the input signal that the subject is trying to reproduce) to be greater in the 1.35 Feedback Gain Condition than in the veridical condition. Recall that the Visual Gain affects the *input* to the visual error computation. We expect the opposite trend for the 0.65 Feedback Gain condition, as is detailed in *Table 3*. Given that we believe the delays are biologically limited based on transmission time of visual information, we expect the delay magnitudes to stay the same between Feedback Gain conditions. Indeed, the model parameters change in the hypothesized manner when different Feedback Gains were applied. The results from fitting the Best 4 (Lowest Err) Model to the data from each of the different Feedback Gains yields the parameter values shown in *Table 4*. The visual feedback delays are nearly identical between Feedback Gain conditions, and the k and Visual Gain values increase and decrease as hypothesized in *Table 3*. The one difference between the hypothesis and these model values is in the lack of a feedback delay for the control subjects. This discrepancy is addressed in the Discussion section.

Lastly, we used bootstrapping to generate confidence intervals for the parameter values in our Best 4 (Lowest Err) Model. Confidence intervals (see Methods) for our best model to be: 2.1–3.4 for $k_{patient}$ and $k_{control}$; 0.32–0.46 for Visual Gain$_{patient}$ and Visual Gain$_{control}$; 174–261 ms for Visual Delay$_{patient}$; and 122–161 ms for Visual Delay$_{control}$, and Feedback Delay$_{patient}$.

**Table 3.** Hypothesized changes in model parameters under different Feedback Gains.
The table shows how we expect the model parameter values indicated in the bottom two rows (1.35 and 0.65 Feedback Gain) to change (with respect to the value obtained in the veridical Feedback Gain condition, Feedback Gain = 1.0) with changing Feedback Gains. We expect Visual Delay of approximately 140 to 250 ms, and Proprioceptive Delay approximately 110 to 150 ms (*Cameron et al., 2014*). We also expect Visual Gain to be less than one in the veridical feedback condition because we do not expect the subjects to attempt to replicate the full magnitude of the signal, given that it is a challenging, fast, and unpredictable task.

| Feedback Gain | K | Visual gain | Visual delay (s) | Feedback Delay$_{patient}$ (s) | Feedback Delay$_{control}$ (s) |
|---|---|---|---|---|---|
| 1.0 | | <1 | Visual Delay | Visual Delay | Proprioceptive Delay |
| 1.35 | Decrease | Increase | Unchanged | Unchanged | Unchanged |
| 0.65 | Increase | Decrease | Unchanged | Unchanged | Unchanged |

**Table 4.** The model (Best 4 Lowest Err) parameters changed with different applied Feedback Gains in a manner consistent with our prediction.

| Feedback Gain | $k_{patient}$ | $k_{control}$ | Visual Gain$_{patient}$ | Visual Gain$_{control}$ | Visual Delay$_{patient}$ (ms) | Visual Delay$_{control}$ (ms) | Feedback Delay$_{patient}$ (ms) | Feedback Delay$_{control}$ (ms) |
|---|---|---|---|---|---|---|---|---|
| 1 | 2.7 | 2.7 | 0.39 | 0.39 | 210 | 141 | 141 | 0 |
| 1.35 | 2.2 | 2.2 | 0.44 | 0.44 | 209 | 140 | 140 | 0 |
| 0.65 | 3.6 | 3.6 | 0.33 | 0.33 | 210 | 144 | 144 | 0 |

## Experiment 2: poor tracking of simple oscillatory trajectories highlights cerebellar patients' predictive deficit

We validated the sinusoidal tracking paradigm by quantifying a known behavioral deficit in cerebellar visuomotor control. In this predictable task, we expected healthy participants would be efficient, exerting less than or equal to the amount of effort required to minimize error for a given phase lag. Where subjects lie relative to this tradeoff can be visualized on a phasor plot; efficient tracking would yield points on or inside the effort/error tradeoff circle (*Figure 2B*, Materials and methods). We hypothesized that patients with impaired prediction may in some cases exert more effort than needed to minimize error, resulting in phasor points outside the effort/error tradeoff circle. We also hypothesized that patients would generate larger tracking errors.

At the five lowest frequencies tested, all control responses landed in/on the circle, whereas only 31% of patient responses landed in/on the circle (*Figure 5*). Additionally, patients generated larger tracking errors: 70% of patient responses exhibited larger error magnitude than the respective age-

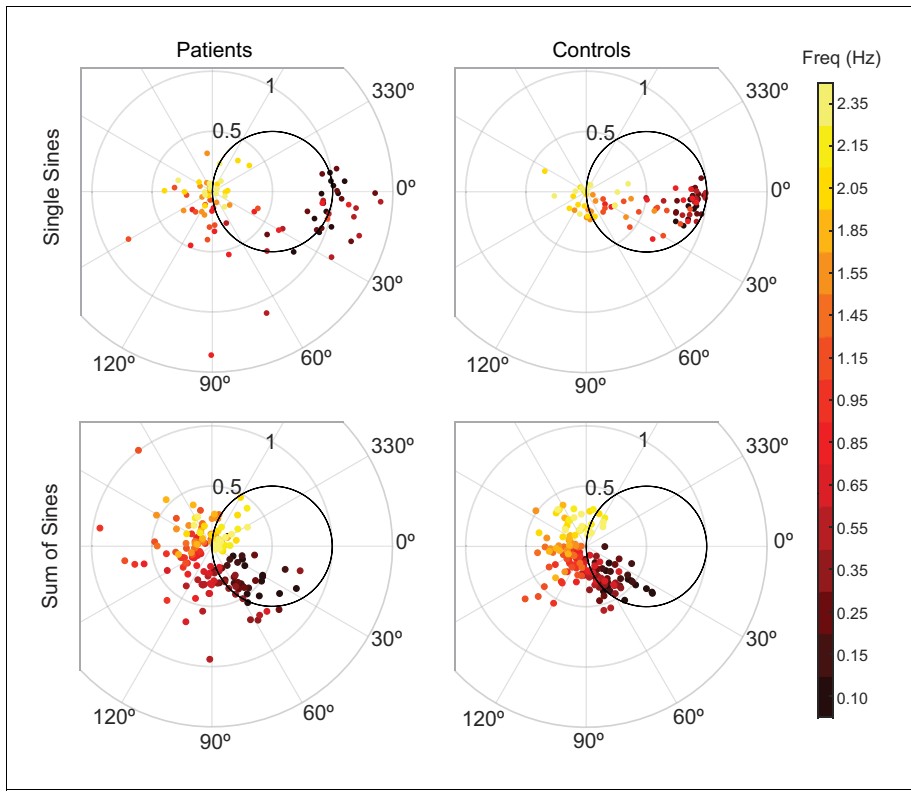

**Figure 5.** Patients and controls have categorically different responses to single-sine and sum-of-sine stimuli. Each data point is the response at a single frequency (magnitude and phase) of a single subject. The patient data in both the single and sum-of-sines conditions are more variable than that of the control subjects. In the single-sines condition, control subjects are able to consistently remain inside the effort/error circle at low frequencies, balancing the tradeoff between effort and error, while patients are not. This result is consistent with previous studies that show patients have impaired prediction.

matched control responses. Furthermore, we created an aggregate tracking error for the single-sines experiment by calculating the sum of the tracking errors for the lowest five frequencies. We compared the aggregate tracking error between patients and age-matched controls by calculating the difference for each pair and testing those differences against 0 using a two-sided sign test. The null hypothesis for a two-sided sign test is that there was no difference between patients and their age-matched controls. The alternative hypothesis is that the controls may be either better or worse than patients. When we compared the difference between the aggregate tracking errors for each patient/control pair to 0, controls were significantly better than patients at single-sines tracking with 10 out of 11 controls performing better than their age-matched counterpart (p=0.012). These results illustrate cerebellar patients' poor predictive ability. Importantly, note that this analysis does not distinguish what portion of their errors stemmed from poor prediction of the stimulus trajectory versus poor control of the arm.

For comparison, we also included the results from the task where the stimulus was unpredictable (sum-of-sines from Experiment 1) in *Figure 5*. For this experiment, because neither group would be able to predict the target trajectory, we expected that patients would be less impaired relative to controls. Indeed, patient and control behaviors were more similar in this condition: for the five lowest frequencies tested in the sum-of-sines condition, 72% of control responses and 64% of patient responses were in/on the circle. Note, however, that there is an increased phase lag of the cerebellar patients (relative to controls), particularly as the frequency of the stimulus increases, which would be expected if the patients were dependent primarily on time-delayed visual feedback (a pure feedback delay introduces greater phase lag at higher frequencies). Specifically, 73% of individual patient frequency responses were more phase lagged than the responses of the age-matched control (as computed on an individual frequency basis).

As expected, we also observed a reduction in tracking error from the sum-of-sines task to the single-sines task. We calculated the magnitude of the tracking error for each subject for each frequency (i.e. for the frequencies that were tested in both conditions) and then compared the errors between the conditions. 93% of controls' and 75% of patients' tracking errors were reduced in the single-sine task in comparison to his or her performance in the sum-of-sines task. Again, we computed the aggregate tracking error for the sum-of-sines task, as described earlier, using the same five frequencies used in the single-sines aggregate tracking error calculation. We then used a two-sided sign test to determine that patients (p=$9\times10^{-4}$) and controls (p=$9\times10^{-4}$) were better at tracking single-sines than sum-of-sines, with all patients and all controls having less tracking error for single-sines than sum-of-sines.

To specifically look at phase lag between groups (patients vs. controls) and conditions (single- vs. sum-of-sines), we used a circular statistical approach. *Figure 6* shows polar representations of the phase lags from the single-sine and sum-of-sines conditions (*Figure 6A,C*) and an example time series from a control and cerebellar subject tracking the 0.85 Hz sine wave (*Figure 6B,D*). The example control subject showed little phase lag suggesting that this individual could make use of an internal prediction of the dot movement and their arm movement. In contrast, the example cerebellar subject showed a systematic phase lag suggesting that their ability to use prediction is impaired. The pattern of lag can be visualized across frequencies in the polar plots. As a group, controls show a small increase in lag as the frequency increases in the single-sine condition, whereas the cerebellar group shows large lags (*Figure 6A and C*, compare purple vectors). Circular ANOVA for the single-sine condition showed that the cerebellar group had greater lags compared to controls (Group effect, p=$1\times10^{-8}$), that the lags increased with frequency (Frequency effect, p=$7\times10^{-11}$) and an interaction such that the cerebellar group lags increased more than controls across frequency (Interaction effect, p=$2\times10^{-6}$). In the sum-of-sines condition, the difference in lags across frequency was qualitatively greater when comparing cerebellar and control groups (*Figure 6A and C*, compare black vectors). Overall the cerebellar group showed greater phase lags compared to controls (Group effect, p=0.02), and there was a statistically significant effect of Frequency (Frequency effect, p=$7\times10^{-30}$). Group x Frequency interaction (Interaction effect, p=0.03), was less significant than that seen in the single-sine condition.

## Experiment 3: virtual reality feedback can reduce dysmetria

The results from Experiment 1 indicated that cerebellar patients' feedback control may be largely intact. In Experiment 3, we leveraged this intact element of their control system to reduce dysmetria.

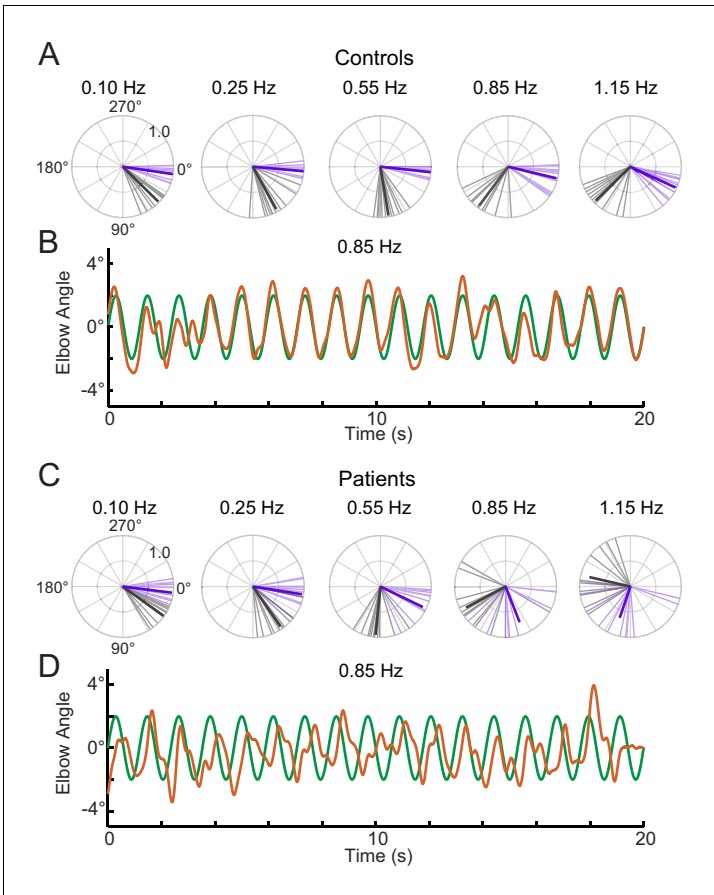

**Figure 6.** Phase lags show different patterns between groups. Polar plot representations showing phase lag for the five lowest frequencies common to the single-sine and sum-of-sines conditions. (A) Control group phase lags. Individual subjects are represented as lighter colored unit vectors and the group mean vector is plotted in bold. Purple indicates the single-sine condition and black represents the sum-of-sines condition. Note that only phase lag is represented on the polar plot (gain is not represented). The Control group is able to track the single-sine stimuli with little or no phase lag, but shows phase lags that increase with frequency in the sum-of-sines condition. (B) Example response from a control subject tracking a 0.85 Hz single-sine stimuli. The target sine wave is represented in green and the subject performance in orange. Note that this control could predict the single-sine and track it with little or no phase lag. (C) Cerebellar group phase lags plotted as in A. The Cerebellar patients were able to track the two slowest single-sine stimuli with little phase lag (purple, 0.10 and 0.25 Hz) but then shows increasing phase lags as a function of frequency. Note that 0.10 and 0.25 Hz are extremely slow frequencies that could be followed using only visual feedback control. In the sum-of-sines condition, the cerebellar group shows phase lags that increase with frequency. (D) Example response from a cerebellar subject tracking a 0.85 Hz single-sine stimuli. In contrast to the control subject, this cerebellar patient shows phase lags relative to the target.

Our goal was to provide each cerebellar patient with customized visual position feedback information that took into account their controller's mismatched feedforward model. This altered visual feedback would help them generate the correct motor command for a simple elbow movement (30° flexion) by accounting for their actual arm dynamics.

Here, we used a motor task that requires subjects make a discrete movement to a target that is stepped to a new position. This was chosen so that we could compare our data to previously published results using the same task with cerebellar patients. Subjects were asked to make a 30° elbow flexion movement from a home target to a stepped target. The task was completed with both veridical visual feedback and with altered visual feedback that was designed to reduce their dysmetria.

Bhanpuri et al. theorized that cerebellar patients have a static mismatch between their controller's internal model of their limb inertia and the actual limb inertia for elbow flexion movements (*Bhanpuri et al., 2014*). Critically, for single-joint elbow flexion or extension, an inertial mismatch

causes an acceleration-dependent error in the internal dynamic model because inertia and acceleration are kinematically yoked. Thus, we predicted that an Acceleration-Dependent Feedback Gain, $k_a$ (block diagram inset in *Figure 7A*) could provide corrective feedback to enhance reaching performance of cerebellar patients. Specifically, we took a subject's estimated acceleration (see Materials and methods), multiplied this by the Feedback Gain, and then added it to the position of the subject's cursor:

$$\text{ShownElbowAngle} = (\text{ActualElbowAngle}) + k_a \cdot (\text{ElbowAngularVelocity})$$

We predicted that positive value for $k_a$ would make a subject more hypermetric and negative value would make a subject more hypometric. Thus, we expected hypometric patients would experience a reduction in dysmetria with a positive gain. Similarly, we expected hypermetric patients would experience a reduction in dysmetria when a negative gain was applied. For a given value for the gain for $k_a$, subjects completed 30° elbow flexion reaching movements in the Kinarm exoskeleton robot. Reaches were categorized as hypometric, hypermetric, or on-target based on the angle where they made their first correction to their movement. The Acceleration-Dependent Feedback Gain $k_a$ was applied to the visual feedback provided on the Kinarm screen. For more details on the implementation and instructions of this task, see Materials and methods.

When an appropriate Acceleration-Dependent Feedback Gain $k_a$ was applied, dysmetria was reduced. The effect of this gain on the displayed trajectory is shown in *Figure 7A*, where the dashed green line shows the altered visual feedback given to the subject and the solid green line shows the subject's actual reach to the target for a single Feedback Gain condition. Increasing the magnitude of $k_a$ had a graded effect on the trajectory profile, as shown by the decreasing amplitude of the solid traces in *Figure 7A*. To find the 'best' $k_a$ we applied the PEST algorithm (see Materials and methods). Applying this gain shifted the angle of first correction in the way we predicted, as shown in both *Figure 7B*. *Figure 7C* shows all the data collected during the implementation of PEST as well as additional trials across a range of gains $k_a$ (see Materials and methods) that were collected for a subset of patients (and all control subjects; see *Figure 8*). Critically, positive Acceleration-Dependent Feedback Gain made patients more hypermetric and negative

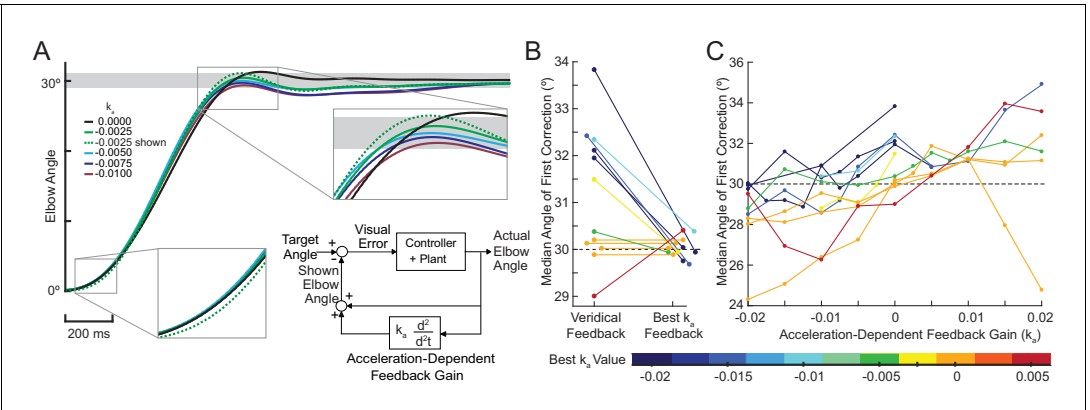

**Figure 7.** Acceleration-dependent feedback reduces dysmetria. (**A**) The trajectories from a single cerebellar patient, moving to the grey target zone. Overshooting or undershooting the grey region is categorized as hypermetric or hypometric, respectively. The solid lines are average trajectories (across trials for a single subject) at a given Acceleration-Dependent Feedback Gain value. A decrease in the angle of first correction (i.e. the peak of the first peak of the elbow angle) is apparent as the Acceleration-Dependent Feedback Gain decreases. The best Acceleration-Dependent Feedback Gain for this subject was determined to be −0.0025, and the resulting movement trajectory is shown as a solid green line. The dashed green line illustrates the trajectory that is displayed to the subject on the screen for the −0.0025 condition. This example also shows that oscillation around the target remains with Acceleration-Dependent Feedback Gain. The block diagram in the lower right shows how the Acceleration-Dependent Feedback Gain was applied to the visual representation of the fingertip position. (**B**) The reduction in dysmetria exhibited by patients when the best Acceleration-Dependent Feedback Gain was applied. Each line represents a single cerebellar patient. Lines are color coded to indicate the best Acceleration-Dependent Feedback Gain for that subject. (**C**) Increasing Acceleration-Dependent Feedback Gain causes increased hypermetria. Again, each line represents a single cerebellar patient. At the highest gain values, the visual feedback diverged enough from the fingertip position so that some subjects paused mid-reach, presumably due to the conflicting visual feedback. This yielded the observed drop in median angle of first correction seen in some subjects in the higher gain conditions. This figure shows only the cerebellar patients. Control subject data is shown in *Figure 8*.

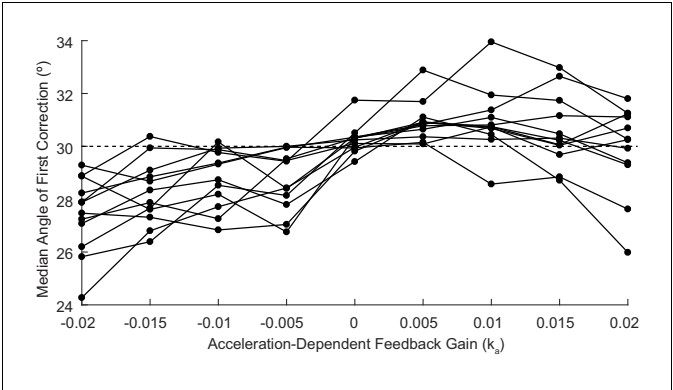

**Figure 8.** Acceleration-Dependent Feedback also affects single reaches in control subjects. Control subjects' median angle of first correction when completing thirty-degree flexion movements. The expected upward trend in median angle of first correction is visible across subjects. At the largest Acceleration-Dependent Feedback Gains, there is a slight drop off in median angle of first correction. At high gain values, the discrepancy between the cursor and the fingertip became more apparent, causing some subjects to pause mid-reach, resulting in a lower median angle of first correction for these high gain conditions.

Acceleration-Dependent Feedback Gain made patients more hypometric; likewise for control subjects. Thus, subject-specific Acceleration-Dependent Feedback Gain was needed to best reduce dysmetria. In general, increased Acceleration-Dependent Feedback Gain magnitude was needed to ameliorate greater dysmetria.

## Discussion

We found that cerebellar patients can use visual feedback control in a manner similar to control subjects. This was assessed during a sum-of-sines visuomotor tracking task by subtly changing the gain of the visual cursor feedback (i.e. hand) and measuring the degree to which an individual subject could modulate the gain of his or her corrective movements. Our modeling corroborated this finding: the structure and parameters between patients and controls were generally the same, except that the patient and control models differed substantively in their feedback delay. The patients relied on time delayed cursor feedback of their hand position, whereas controls appeared to be able to generate predictions of their hand position.

We also saw increases in phase lag for patients versus controls when following predictable sine waves. While patients showed large phase lags that increased with frequency in both conditions, controls only showed large phase lags in the sum-of-sines condition when the stimulus was unpredictable. We interpret this to mean that the control subjects could predict the movement of *both* the hand *and* stimulus in the single-sine condition, but the cerebellar patients could not predict *either* as reliably as controls, relying more heavily on visual, moment-to-moment feedback for tracking.

The above results are significant as it has previously been difficult to assess visual feedback control in these patients due to their clear impairments in feedforward control of movement. In Experiment 3, we tested whether patients' largely intact visual feedback could be exploited to benefit their targeting performance in a reaching task. Indeed, we found that cerebellar patients could leverage their intact, albeit delayed, feedback control ability to improve their movement when we manipulate their visual feedback to assist them.

### Delay in patients but not in healthy controls

Cerebellar patients are thought to make faulty state estimates of arm motion due to a damaged internal model, which adversely affects their proprioceptive estimates during active movement (*Bhanpuri et al., 2013*). Thus, we expect them to increase their reliance on visual feedback, which is time delayed compared to proprioceptive feedback. Here we hypothesized that our patients' feedback control system would have an increased delay in comparison to controls. This hypothesis is

supported by work that showed that healthy subjects normally rely more on proprioception than vision for feedback control (*Crevecoeur et al., 2016*). Based on the *Crevecoeur et al., 2016* model, increased variability of the proprioceptive information would lead to increased weighting of visual feedback (*Crevecoeur et al., 2016*; *Weeks et al., 2017*), which comes at the cost of increased delay for patients compared to controls.

Our modeling results reveal that patients have a delay in their feedback loop that approximates what one would expect from visual feedback, while controls appear to have negligible feedback delay. One interpretation of the control data is that controls are relying on a predictive internal model of their arm dynamics. This would allow them to make corrections even faster than they could using purely proprioceptive feedback. In other words, healthy controls can make corrections for current target movements based on a prediction of the present limb state from outdated measurements and known motor commands, rather than waiting on delayed sensory feedback.

We think that the cerebellar patients' deficits are best explained by their dependence on time-delayed visual feedback, though it is theoretically possible that delays in motor execution could contribute. Previous studies have shown that some patients with cerebellar damage exhibit longer than usual reaction time delays when asked to make discrete, rapid movements (*Beppu et al., 1987*; *Beppu et al., 1984*; *Day et al., 1998*; *Holmes, 1917*; *Vilis and Hore, 1980*). These reaction time studies aimed to quantify a movement initiation delay as opposed to a feedback delay. Here, we specifically assessed the timing of the in-flight trajectory alterations made while tracking an unpredictable stimulus. While our feedback delay is a type of 'reaction time' because this task requires making corrections on the fly, the neural mechanisms underlying these corrections may differ from those responsible for planning and initiation of a rapid movement.

Cerebellar patients can also show a reduced feedback gain in situations where responses are driven by proprioception more than vision. *Kurtzer et al., 2013* studied cerebellar patients' ability to modulate long latency stretch responses (i.e. EMG responses within 45–100 ms of a muscle stretch) that depend on knowledge of intersegmental dynamics of the arm. This was assessed by mechanically perturbing the elbow joint and measuring long latency responses from elbow muscles and from shoulder muscles in anticipation of the passive shoulder motion induced by elbow movement. Cerebellar patient responses were found to have a lower gain and a slight delay in timing compared with controls. We interpret this in the context of the different task demands between studies– in that study, proprioception drove the response; here vision appears to be more important to patients. These results combined suggest that cerebellar subjects can reweight which feedback modality to rely on (proprioception versus vision) depending on task demands, consistent with *Block and Bastian, 2012*, and may reduce the sensorimotor gain on either modality to minimize the negative effects of delay.

Lastly, while the confluence of data and modeling presented here seems to indicate that patients rely disproportionately on delayed feedback from vision for this visuomotor tracking task, two of our top models (Model Best 5, Lowest Err and Best 4, Lowest Err; see *Table 2*) actually exhibited shorter feedback delays than visual delays for patients. This suggests that the patients' control systems in this task may include some reliance, albeit weaker, on prediction and proprioception.

## Modeling results support the cerebellum as a smith predictor

Sensorimotor tasks are naturally analyzed using control systems theory and system identification (*Roth et al., 2014*), and because cerebellar patients' dysmetria resembles a poorly tuned control system, many researchers have analyzed cerebellar behavior using control theory (*Luque et al., 2011*; *Manto, 2009*; *Miall et al., 1993*; *Porrill et al., 2013*).

Our modeling results indicate that control subjects make corrections based on prediction of a future limb state measurement. While there is some controversy on this (*Pélisson et al., 1986*; *Wolpert et al., 1998*), previous work has proposed that the cerebellum might function in a manner consistent with a Smith Predictor (*Miall et al., 1993*). Essentially, a Smith Predictor is a control architecture that compensates for self-movement feedback delay. To achieve this requires a simulation of the plant in order to predict sensory consequences of motor output, and a simulated delay that 'stores' the plant simulation for comparison with delayed feedback. In order to understand the consequences of incorporating a Smith Predictor, consider the generic feedback block diagram of sensorimotor control depicted in *Figure 9* (top) and model the brain as either a controller *with* (*Figure 9*, left) or *without* (*Figure 9*, right) a Smith Predictor. Interestingly, upon simplification, we

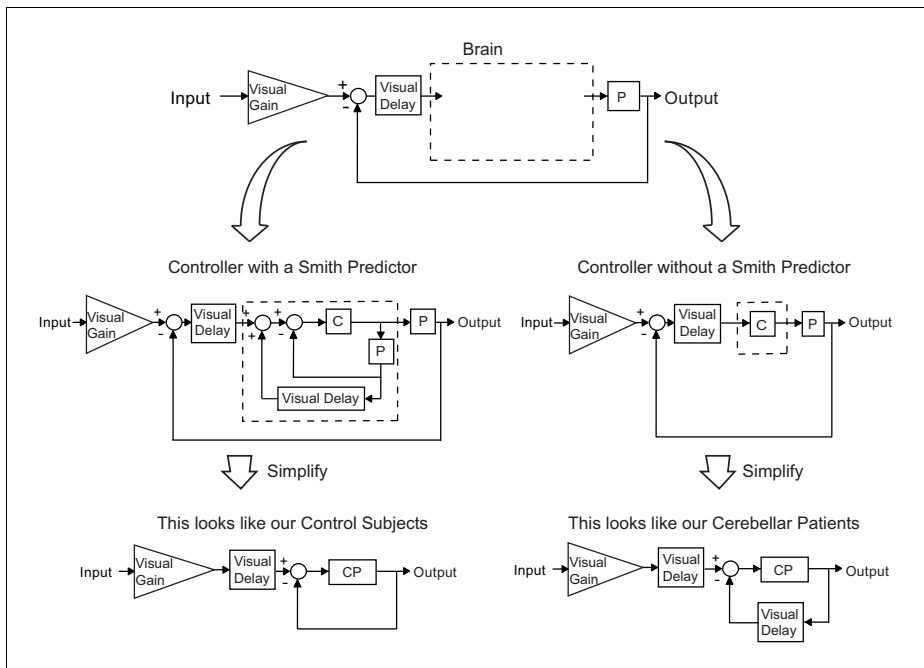

**Figure 9.** Our modeling results agree with the theory of the cerebellum as a Smith Predictor. A basic model for the control system is shown in the top panel. In our experiment, the plant, P, represents the entire mechanical arm system, including the robot exoskeleton arm. A single visual delay block delays the input from both the reference signal and the feedback information. A visual gain scales the amount of the input the subject will attempt to replicate. The brain, illustrated by a dashed box, can be modeled as containing just a controller, C, as shown on the right side. Or, the brain can be modeled as a controller with an accompanying Smith Predictor, as shown on the left side. When these two alternate structures are simplified, we are left with the exact model structures yielded by our previous modeling results.

find the exact structures yielded by two of the top models (*Best 5 Lowest Var* and *Best 4 Lowest Var*): the patient model matches the simplified block diagram without a Smith Predictor and the age-matched control model matches the simplified block diagram which one. This suggests that an intact cerebellum may act in a manner qualitatively similar to a Smith Predictor and that patients ability to perform such prediction is impaired.1.

Perhaps the increased feedback delay in the cerebellar patients is unavoidable if one assumes the patients' control systems lack the essential capabilities of a Smith Predictor (making and storing predictive plant simulations). Thus, the apparent increased reliance on visual feedback seen in cerebellar patients may not be a compensatory mechanism at all—perhaps all subjects rely on delayed visual feedback, but healthy subjects disguise the delay using predictions of future measurements of limb state. Patients may simply lack this capability and it is not that they *compensate* for their deficit by relying *more* on visual feedback but instead that the loss of the sensory prediction and storage reveals the visual feedback delay inescapably present in both groups. This would provide some clarity on how cerebellar patients could 'learn' to 'compensate' in this way even when their cerebellum was damaged.

At its core, the Smith Predictor has to do with simulating the mechanical plant (and appropriate delay) in order to mitigate the effect of *feedback* delay. But notice that in our model there is a large visuomotor delay on the reference input even with the Smith Predictor in place. In order to compensate for the delayed visual measurement of the stimulus, the brain would need to be able to predict reference motions. To avoid this confound of stimulus prediction, and focus on feedback delay compensation, we fit our models using responses to unpredictable (sum-of-sine) target motions.

But, what if the target motion were predictable? Previously, it was shown in another species (weakly electric fish) that target predictability can indeed improve tracking performance, and it was hypothesized that this was based on an internal model of predictable reference motion (*Roth et al., 2011*). Here, we extend this to healthy human subjects who show marked improvement in the

single-sine (predictable) tracking task. Cerebellar patients exhibit much more modest improvement in tracking predictable stimuli, suggesting that they may have impairment in both their own plant model predictions and predictions of external sensory references (see Results, Experiment 2). Further data and experiments would be needed to better isolate the effects of cerebellar damage on these potentially disparate functions of the cerebellum.

Lastly, even if a healthy cerebellum helps 'cancel' expected cursor feedback, this does not imply that visual cursor feedback goes unused: any errors in visual feedback, i.e., cursor self-motion feedback that is not precisely anticipated by the cerebellum, could be highly informative, and recent evidence (*Yon et al., 2018*) suggests that a predictive model of action–perception could heighten perceptual sensitivity, making self-motion (in this case, cursor) feedback more nuanced and precise.

There are potential alterative modeling interpretations of the data presented in this paper. For example, a state predictor with optimal state feedback (*Crevecoeur and Gevers, 2019*) is consistent with our observation that healthy subjects compensate for delay and is an avenue for further investigation. Also, our observation that visual target delay was longer in the patient model than the control-subject could conceivably be due to a different source of phase lag that our simplified modeling structure was not able to capture. Future experiments and analysis will investigate this further.

## Potential for real-world application

In Experiment 3, we were successfully able to reduce the dysmetria of both the hypermetric and hypometric cerebellar patients in an elbow flexion task. Our visual feedback-control-based intervention was developed based on a control model which represented dysmetria as a mismatch between the patient's internal model of their limb inertia and the actual limb inertia (*Bhanpuri et al., 2014*). Thus, the success of our intervention provides support to this inertial mismatch theory. However, recent evidence based on the application of limb weights suggest that the inertial mismatch theory may be insufficient to explain dysmetria in multi-joint movements (*Zimmet et al., 2019*). Thus, feedback enhancement of multi-joint movements will likely require modifications of the simple acceleration-based controller tested in this paper. Moreover, the 'best' acceleration-dependent feedback gain was patient specific even for the single-degree-of-freedom reaches tested for this study, suggesting that idiosyncrasies in each patient's motor control system (*Kuling et al., 2017*; *Rincon-Gonzalez et al., 2011*) may mandate patient-specific tuning. Such tuning will be made more complex for 2D and 3D arm movements.

Though our intervention successfully reduces the initial over or undershooting component of the reach, our method does not attempt to correct all aspects of dysmetria. Patients with cerebellar ataxia also typically exhibit increased movement variability. Application of this visual feedback gain does not attempt to reduce the variability of patient reaches. Secondly, our intervention does not completely eliminate the oscillations experienced by the subject after this first correction is made. Moreover, our modeling work does not characterize or explain this increased variability directly. Mild oscillations after the first correction can be seen in *Figure 7A*. It is possible that adding a damping term to the feedback intervention might be able to address this oscillatory behavior.

Our results provide an encouraging foundation for future studies because they show that cerebellar patients are capable of using their own intact visual feedback control system to make accurate reaches. Further study is needed to determine whether this intact feedback control system could be leveraged therapeutically to reduce dysmetria without need for a virtual reality system. Alternatively, it is possible that wearable sensory augmentation (for example, tendon vibration or skin stretch) could be used as a surrogate for the vision-based sensory shift provided here. These alternatives are especially important because drug therapy for cerebellar ataxia is not currently viable, leaving rehabilitative and assistive therapies alone as the primary means of treatment for these patients. In any case, we hope that by identifying these intact movement control mechanisms we might help move treatment possibilities forward.

## Acknowledgements

National Institute of Child Health and Human Development HD040289 to AJB and NJC, Applied Physics Lab Graduate Fellowship FNACCCX19 to AMZ, National Science Foundation 1825489 to NJC and AJB. Thanks to James Freudenberg, Brent Gillespie, Adrian Haith, Matthew Statton, and

Sarah Stamper for helpful comments and discussion. Thanks to Amanda Therrien for recruitment of patients, clinical assessments, and feedback throughout this process.

## Additional information

### Competing interests
Noah J Cowan: Reviewing editor, *eLife*. The other authors declare that no competing interests exist.

### Funding

| Funder | Grant reference number | Author |
|---|---|---|
| National Institutes of Health | HD040289 | Amy J Bastian<br>Noah J Cowan |
| Applied Physics Laboratory Graduate Fellowship | FNACCCX19 | Amanda M Zimmet |
| National Science Foundation | 1825489 | Di Cao<br>Amy J Bastian<br>Noah J Cowan |

The funders had no role in study design, data collection and interpretation, or the decision to submit the work for publication.

### Author contributions
Amanda M Zimmet, Conceptualization, Data curation, Software, Formal analysis, Funding acquisition, Investigation, Visualization, Methodology, Writing - original draft, Writing - review and editing; Di Cao, Data curation, Software, Formal analysis, Investigation, Visualization, Methodology, Writing - review and editing; Amy J Bastian, Conceptualization, Resources, Data curation, Software, Formal analysis, Supervision, Funding acquisition, Investigation, Visualization, Methodology, Project administration, Writing - review and editing; Noah J Cowan, Conceptualization, Data curation, Software, Formal analysis, Supervision, Funding acquisition, Investigation, Visualization, Methodology, Writing - original draft, Project administration, Writing - review and editing

### Author ORCIDs
Amanda M Zimmet http://orcid.org/0000-0003-1457-3072
Di Cao http://orcid.org/0000-0002-1547-9929
Amy J Bastian https://orcid.org/0000-0001-6079-0997
Noah J Cowan https://orcid.org/0000-0003-2502-3770

### Ethics
Human subjects: The experimental protocol was approved by the Institutional Review Board at Johns Hopkins University School of Medicine (protocol # IRB00182673) and all participants gave informed consent prior to joining this study, according to the Declaration of Helsinki.

### Decision letter and Author response
Decision letter https://doi.org/10.7554/eLife.53246.sa1
Author response https://doi.org/10.7554/eLife.53246.sa2

## Additional files

### Supplementary files
• Transparent reporting form

## Data availability

All data generated or analyzed during this study is openly on the JHU Data Archive under https://doi.org/10.7281/T1/BCARLC. The data is available here: https://archive.data.jhu.edu/dataverse/LIMBS.

The following dataset was generated:

| Author(s) | Year | Dataset title | Dataset URL | Database and Identifier |
|---|---|---|---|---|
| Zimmet AM, Cao D, Bastian AJ, Cowan NJ | 2020 | Data Associated with publication: Cerebellar patients have intact feedback control that can be leveraged to improve reaching | https://doi.org/10.7281/T1/BCARLC | JHU Data Archive , 10.7281/T1/BCARLC |

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
