## [Decision Letter]

**Acceptance summary:**

Skilled movement involves the interplay of feedforward and feedback processes. A large body of evidence has provided compelling evidence of the critical role for the cerebellum in feedforward control. In contrast, cerebellar involvement in feedback control has been more controversial. In this study, Zimmet and colleagues show that individuals with cerebellar ataxia are able to use visual feedback in a similar manner as control participants. However, unlike the controls, they were unable to compensate for delays in feedback processing-thus, their inability for predictive control resulted in delayed feedback. Interestingly, if the feedback was phase advanced in a virtual reality system, the patients' performance improved, providing a nice demonstration of the interplay between feedback and feedforward processes.

**Decision letter after peer review:**

Thank you for submitting your article "Cerebellar patients have intact feedback control that can be leveraged to improve reaching" for consideration by *eLife*. Your article has been reviewed by three peer reviewers, and the evaluation has been overseen by Richard Ivry as Reviewing Editor and Senior Editor. The following individuals involved in review of your submission have agreed to reveal their identity: Max Donelan (Reviewer #2); Frederic Crevecoeur (Reviewer #3).

The reviewers have discussed the reviews with one another and the Reviewing Editor has drafted this decision to help you prepare a revised submission.

Summary:

The manuscript investigates the role of the cerebellum in feedback control, comparing the performance of individuals with cerebellar pathology to a matched control group. The patients failed to exhibit predictive control (feedforward control), instead showing a strong dependency on the visual feedback (feedback control). The authors simplified a complicated control system intro tractable models, evaluated the model across different topologies, and designed experiments to distinguish between feedback control in the absence of prediction, and feedback control that leverages prediction. Their findings support the hypothesis that while, control participants could combine proprioceptive feedback with an internal model to have a prediction of the body's state and/or the target motion to allow for zero-delay tracking, the patients with cerebellar damage could not. The final experiment provides a strong and courageous test of these ideas, using a manipulation that partly corrected for the patients' deficiencies by shifting the visual information in an idiosyncratic manner to optimize predictive control to the visual feedback. The manuscript is clearly of great interest as it provides valuable data and insight with respect to cerebellar-dependent motor impairments.

Essential revisions:

1) Issues with Experiment 3: Is the experimental manipulation about the manipulation of the delay, or does it become one of a manipulation of gain? If you would add acceleration feedback to a response that consists of sines (as in experiments 1 and 2), you add a signal that is 180 degrees out of phase, which simply leads to a reduction of gains (not a correction for delays as claimed, see the Introduction, last paragraph). For single point-to-point movements, adding acceleration could make sense, but does this make a clear prediction relevant to the goals of the current study? It seems as if it also here acts as a change of gain. The authors show that the optimal amount of acceleration (and thus the gain) is idiosyncratic. This corresponds with idiosyncratic biases in visuomotor matching that have been reported in the literature (e.g. in Rincon-Gonzalez et al.,(011, and Kuling et al., 2017). Gain control may be in intact in the patients (or experiment may lack sensitivity to detect gain control given ease in learning these).

2) Complexity of the analyses. These analyses are smart but difficult to follow and there may be alternatives that either simplify the presentation or are better suited to the questions under consideration.

a) The authors argue that the use of visual feedback introduces a delay. Instead of assessing the delay, they present the results in terms of a frequency-dependent phase-lag (illustrated by phasor plots that require some explanation). It may be simpler to regard the observed phase lags as corresponding to a delay of a bit more than 100 ms for al frequencies. The same argument holds for switching between the frequency domain and spatial analyses.

b) The models would benefit from more explanation. In particular, it requires a big leap of faith from the reader to be willing to accept that the controller and plant can be modeled as a simple integrator. It required in part, reading the McRuer and Krendel, 1974 paper. But requiring readers to seek it out, read it, and understand it gets in the way of the digesting the present manuscript.

c) It took a while to understand why the acceleration term is like prediction. Others may benefit from a clear explanation in the Introduction that builds their intuition.

d) The manner in which you chose the appropriate gain on the acceleration term may suffer from post-hoc selection bias. This concern was ameliorated by Figure 7C and Figure 8 which show clear systematic effects of this gain on dysmetria.

e) The developments are based on a simple model which ignores much of recent work on feedback control and delay compensation: state-of-the-art control models consider state-feedback control (Todorov and Jordan, 2002), and a lot of work has been done to characterise feedback functions and their respective latencies (Scott, Trends in neurosciences 39 (8), 512-526). Delay compensation has been studied in this context following visual perturbations (Izawa and Shadmehr, 2008) and mechanical perturbations (Crevecoeur and Scott, 2013). Recently it was proposed that in theory, a faulty delay compensation within the stability margin can produce oscillations in eye and arm movements, but these two operations are needed in both visual and limb motor systems (Crevecoeur and Gevers, Neural computation 31 (4), 738-764). While recognizing all models have limitations, the authors should consider this framework. The reason is that this literature has found consistent delays of ~100-120ms for the visual system, and ~50-60ms for somatosensory feedback. These are physiological measurements so if there is no reason to suspect a change in conduction velocity for cerebellar patients, these numbers could be used in in a state-feedback control model. This concern is based on the fact that the calculated delays by fitting the McRuer integrator with a Smith predictor seem off (170ms). If the model structure does not correspond to the physiology, the fitted parameters can be precise but inaccurate. We realize that considering state-feedback control model and delay compensation in the framework of LQG is more complex. However, this may be justified as this framework has been used extensively to model reach control and feedback responses to mechanical and visual perturbations.

f) The previous study by Kurtzer on long-latency control in cerebellar patients (Kurtzer et al., 2013) was not used in this study since it did highlight a reduction in response gain specific to the long-latency motor system, which is a valuable data for modelling purposes.

3) The results show that the patients lack delay compensation for the predictable visual stimulus. The question arises as whether the problem was delay compensation with respect to their body movements or with respect to the external world. The behavior of Exp 1 is used to conclude that the patients had impairments in their estimate of limb state. This conclusion seems indirect and it is not based on a limb perturbation. The authors imply that delay compensation is associated with proprioceptive estimates only, and that it is not expected to occur in the visual system. The associated rationale is not clear. It would seem that compensation for delays in the visual system is also necessary for controls to track the predictable target. Can we conclude that feedback control is intact, or is this specific form of visuomotor control that is unaffected? Possibly feedback control in the proprioceptive system is also intact, what seems to be missing is delay compensation, but it was not demonstrated whether the faulty delay compensation was in the visual or somatosensory motor system. The lack of predictive control could be in the visual system based on the tracking experiments. The deficit was attributed to predictive control of the limb based on indirect fitting of the model, but the rationale for this is unclear.

Related to this issue, the theoretical background suggests a role of the cerebellum in predicting the behavior of one's body (internal model). It is unclear how the experiments test this idea. Experiments 1 and 2 differ in the predictability of the stimulus, not the predictability of the body. The different behavior of the cerebellar patients in experiment 2 (and not in experiment 1) would indicate a problem with the prediction of the world, rather than of the body.

4) It was suggested that cerebellar patients had intact feedback control based on experiment 1. This is likely an overstatement: qualitative differences were visible in the traces of Figure 4 and the interaction between conditions and groups was associated with p = 0.07. Thus, it could be that the relatively low sample size does not allow to conclude for differences based on the analyses suggested.

---

## [Author Response]

Essential revisions:1) Issues with Experiment 3: Is the experimental manipulation about the manipulation of the delay, or does it become one of a manipulation of gain? If you would add acceleration feedback to a response that consists of sines (as in experiments 1 and 2), you add a signal that is 180 degrees out of phase, which simply leads to a reduction of gains (not a correction for delays as claimed, see the Introduction, last paragraph).

This question raises a fairly complicated way to look at acceleration-dependent feedback, but that may have arisen from how we wrote the manuscript. Acceleration-dependent feedback leads position feedback by 180 degrees, but the amplitude scales with temporal frequency. This is scaled and added to position feedback, providing predictive feedback. In practical terms, the acceleration-dependent feedback is based on a moving-average estimate of acceleration, which amounts to including a finite-impulse-response (“sliding window”) low-pass filter to the acceleration computation. We used N=100 data points and a sampling period of 1ms (1kHz sampling); based on this, one can develop a simple approximation to our acceleration dependent feedback, showing in Matlab notation, as follows:

s = zpk('s');

Ts = 1e-3; % sampling time

N = 100; % num samples

MovAvg = (1/N)*(1-exp(-s*Ts*N))/(1 – exp(-s*Ts)); % Laplace transform for a 100ms moving average ka = 0.02; % example acceleration gain used in our experiments w = 2*pi*logspace(-1,log10(2.5),50); options = bodeoptions;

options.FreqUnits = 'Hz'; % bode( 1 + ka * s^2 * MovAvg,w,options);

The Bode plot shown in Author response image 1 (of just the combination of position and acceleration-dependent feedback) shows that this feedback is basically a high-pass (phase leading) filter, as expected:

When the gain ka is less than zero, this becomes a lag filter. The implementation of this filter and the practical details are now better described in the Materials and methods (subsection “Acceleration-Dependent Feedback (Experiment 3)”).

For single point-to-point movements, adding acceleration could make sense, but does this make a clear prediction relevant to the goals of the current study? It seems as if it also here acts as a change of gain. The authors show that the optimal amount of acceleration (and thus the gain) is idiosyncratic. This corresponds with idiosyncratic biases in visuomotor matching that have been reported in the literature (e.g. in Rincon-Gonzalez et al.,(011, and Kuling et al., 2017). Gain control may be in intact in the patients (or experiment may lack sensitivity to detect gain control given ease in learning these).

First, we want to clarify, in light of our response above, that acceleration gain and a position gain are not the same thing. That said, we agree that gain control may be intact in these patients and indeed our Experiment 1 demonstrates this (see Figure 3C). In that experiment, we increased (decreased) the overall *position* feedback gain and patients and age-matched controls responded similarly, both decreasing (increasing, respectively) their response in a similar manner.

Also, the apparently idiosyncratic response to acceleration is actually quite consistent across patients in the sense that while they all have their own “optimal” value, each patient tends to trend in a similar way with respect to a change in acceleration-dependent feedback gain. See Figure 7C, for example, where all patients show a positive slope in their Mean Angle of First Correction as a function of *k_a_*. The “optimal” is different because of each patient’s idiosyncratic baseline behavior, giving rise to a correspondingly idiosyncratic “intercept” in Figure 7C.

Thank you for pointing out these references. We had a bit of trouble finding an accurate way to map Kuling et al., 2017, to our current study. Rincon-Gonzalez et al., 2011, shows the proprioceptive map of the arm is systematic and stable, but idiosyncratic. Hand position (unseen) estimation was mapped under conditions with and without tactile feedback and different hands. Each subject had a unique pattern of errors that was stable between hands and over time. This may be related to why the gain (ka) of acceleration is idiosyncratic, trying to compensate for the idiosyncratic estimation error. We adjusted the acceleration feedback by changing gain Ka, which in fact adjusts the filter we add externally, as you can see in the following figure:

**Author response image 2. respfig2:** 

While the relationship between the Rincon-Gonzalez paper and the present manuscript is not completely clear, it does seem that the idiosyncratic nature of the mappings could be related, so in the Discussion, we have added the following general sentence to that effect:“Moreover, the “best” acceleration-dependent feedback gain was patient specific even for the single-degree-of-freedom reaches tested for this study, suggesting that idiosyncrasies in each patient’s motor control system (Rincon-Gonzalez, Buneo, and Tillery, 2011; Kuling et al., 2017) may mandate patient-specific tuning. Such tuning will be made more complex for 2D and 3D arm movements.”

2) Complexity of the analyses. These analyses are smart but difficult to follow and there may be alternatives that either simplify the presentation or are better suited to the questions under consideration.a) The authors argue that the use of visual feedback introduces a delay. Instead of assessing the delay, they present the results in terms of a frequency-dependent phase-lag (illustrated by phasor plots that require some explanation). It may be simpler to regard the observed phase lags as corresponding to a delay of a bit more than 100 ms for al frequencies. The same argument holds for switching between the frequency domain and spatial analyses.

The phasor plot is used to show the detailed frequency responses of all control and patient individuals, then reveal the categorical differences. In the modeling part, which is aimed at capturing the main group differences of the internal model structure, we did consider the pure delay model structure, a pure delay multiplied by a pure gain (Figure 4—figure supplement 1B, Pure Delay model structures (D1, D2, D3)). Further, you can see in the Figure 4—figure supplement 1C and D, although the pure delay models show good model consistency, they all present high fit error in both frequency and time domain, indicating that they do a poor job of capturing the data.

While we agree that the vast majority of the differences in phase lag between the populations can be accounted for by the delay, phase lag by itself is not equivalent to delay. Indeed, even in our extremely simplified model, as the reviewers rightly point out, we wound up with a delay that was probably on the high side (170ms), i.e. our model fits were consistent but biased (we agree with this, more later). Our model includes feedforward (measuring the target) and feedback (measuring self-motion) delays, as well as an additional 90deg phase lag in the feedback loop thanks to the McRuer-style integrator for the Plant + Controller. So it is not quite accurate to try to capture all the phase lag with a delay as this will miss several important factors:

1) Low-pass muscle and mechanical dynamics, which we have coarsely tried to capture with the integrator in the feedback loop

2) It would not discern between lags produced by self-motion feedback (which can be un-delayed due to model-based self-motion prediction in healthy subjects) and feedforward measure delay of the target (which cannot be eliminated unless the target is predictable).

Our model is simple and thus doesn’t capture everything perfectly either, and we elaborate on this below, but simplifying everything to just a delay would undermine our ability to make sensible comparisons about the structural differences between the populations.

b) The models would benefit from more explanation. In particular, it requires a big leap of faith from the reader to be willing to accept that the controller and plant can be modeled as a simple integrator. It required in part, reading the McRuer and Krendel, 1974 paper. But requiring readers to seek it out, read it, and understand it gets in the way of the digesting the present manuscript.

We agree that this literature will not be familiar to everyone and we further improved the writing by adding more explanations of McRuer gain-crossover model in the Materials and methods (subsection “Modeling (Using Data from Experiment 1)”).

c) It took a while to understand why the acceleration term is like prediction. Others may benefit from a clear explanation in the Introduction that builds their intuition.

Agreed; some intuitive explanation was added for the Acceleration-Dependent Feedback in the Materials and methods (subsection “Acceleration-Dependent Feedback (Experiment 3)”). Also more details are shown in our reply to question 1 above.

d) The manner in which you chose the appropriate gain on the acceleration term may suffer from post-hoc selection bias. This concern was ameliorated by Figure 7C and Figure 8 which show clear systematic effects of this gain on dysmetria.

We believe our lack of clarity here caused a misunderstanding. We performed two sets of experiments:

First, we used the PEST algorithm, an iterative experimental method for finding an optimal parameter based on a psychometric evaluation at each step of the algorithm. Specifically, cerebellar patients performed blocks of 10 reaches where this gain was held constant. Based on the categorization for the 10 reaches (hypometric, on-target, or hypermetric), the gain was increased, remained the same, or decreased, as determined by the PEST algorithm (see text for details). We then applied the final gain and measured the angle of first correction using that value of *k_a_* for 10 reaches.

Second, we performed an entirely independent experiment where we tested several gains *k_a_*, and demonstrated a strong positive correlation across those patients that were tested (not all patients could be tested in this way, see text).

All of the data from both experiments are plotted in Figure 7C. We do not believe there is any serious concern of post-hoc selection bias.

To help clarify this, we completely rewrote the Materials and methods section describing these experiments (—subsection “Acceleration-Dependent Feedback (Experiment 3)”) and we’ve clarified the Results (subsection “Experiment 3: Virtual Reality Feedback Can Reduce Dysmetria”).

e) The developments are based on a simple model which ignores much of recent work on feedback control and delay compensation: state-of-the-art control models consider state-feedback control (Todorov and Jordan, 2002), and a lot of work has been done to characterise feedback functions and their respective latencies (Scott, Trends in neurosciences 39 (8), 512-526). Delay compensation has been studied in this context following visual perturbations (Izawa and Shadmehr, 2008) and mechanical perturbations (Crevecoeur and Scott, 2013). Recently it was proposed that in theory, a faulty delay compensation within the stability margin can produce oscillations in eye and arm movements, but these two operations are needed in both visual and limb motor systems (Crevecoeur and Gevers, Neural computation 31 (4), 738-764). While recognizing all models have limitations, the authors should consider this framework. The reason is that this literature has found consistent delays of ~100-120ms for the visual system, and ~50-60ms for somatosensory feedback. These are physiological measurements so if there is no reason to suspect a change in conduction velocity for cerebellar patients, these numbers could be used in in a state-feedback control model. This concern is based on the fact that the calculated delays by fitting the McRuer integrator with a Smith predictor seem off (170ms). If the model structure does not correspond to the physiology, the fitted parameters can be precise but inaccurate. We realize that considering state-feedback control model and delay compensation in the framework of LQG is more complex. However, this may be justified as this framework has been used extensively to model reach control and feedback responses to mechanical and visual perturbations.

We see that there are two main questions here, one is the issue of delay, the other is the use (or lack thereof) of state space modeling.

Delay:

As mentioned above, we agree that our initial estimate of delay of 170ms was long, and this arose due to parameter bias (as described in detail, below). From the literature, estimates vary a bit, but are generally in the range of 110 – 160ms:

Haith, Pakpoor, and Krakauer, 2016: 143ms (mean online visuomotor feedback latency)

Pruszynski, Johansson and Flanagan, 2016: 110 ms (target jump, vision to kinematics)

Franklin and Wolpert, 2008: 151.5 ms (target jump, vision to force)

Saunders and Knill, 2003: 163 ms (online correction to self-motion perturbation during reaching)

Day and Lyon, 2000: 125 ms – 160 ms (target jump, vision to kinematics)

Brenner and Smeets, 1996: 110 ms (target jump, vision to kinematics)

We have updated our text reviewing visuomotor delays from the literature (subsection “Cerebellar Patient Performance Best Captured by Long Latency Closed-Loop Model”), citing delays of 110-160ms as reported above.

Discussion of delay parameter bias in our original manuscript. Initially, for our modeling, we had hand-picked a fairly large (63) subset of all possible models of the general model structure in Figure 4A. However, we were short sighted and left out critical model configurations that allowed more appropriate flexibility in model parameters. For this revision, we modified the code to test all possible 2667 model configurations and discovered models that were better in error and inconsistency as described in the paper. While these improved models are highly similar to the “best models” we had proposed before, there were some key differences, especially in regards to delay. For example, we have found visual tracking delay in healthy controls around 141-147ms, which is consistent with the above literature. Surprisingly (to us, although perhaps not in hindsight) patients had considerably longer delays (181-211ms) when we allowed the parameters to vary in the right way. Our short-sightedness stemmed from now allowing this “de-yoking” of the delays in the right way, essentially “blending” the delays of patients and controls, winding up with a model that was a compromise. Classic example of parameter bias!

So it seems likely that the initial “bias” that this reviewer so poignantly anticipated was caused by an unintentional, but initially forced compromise between patients and controls. Indeed it is an interesting result that the best model configurations we have now discovered exhibit an increase in delay of about 60 ms give or take for patients, a result consistent with observed delay differences in movement initiation.

This modeling correction has led to several small changes in the manuscript (mostly in the subsection “Cerebellar Patient Performance Best Captured by Long Latency Closed-Loop Model”), including a reworking of Figure 4 (including supplementary figures), and updated nomenclature for the models throughout the manuscript. The most scientifically significant component of this is the difference in delay between patients and controls. This is discussed in the aforementioned subsection.

State space:

Since receiving this review, Dr. Cowan and his PhD student Di Cao have begun working extensively with our collaborator James Freudenberg at Michigan to connect McRuer + Smith Predictor like computations to Optimal Feedback Control Theory (OFCT). The new analyses that these reviews have prompted are extremely interesting but far from tidy, and including them would greatly complicate an already complicated manuscript. We are taking them very seriously but feel that including the math here would make the paper (which ultimately is most useful for its experiments and data analysis) impenetrable and very long. We are working very fast to make this all the subject of an entire paper that we believe will be readable by computational motor control readers. We will be sure to acknowledge the reviewers of this paper that have disclosed themselves, and tie it to the present manuscript, if published.

Now, as to the accessibility of the models we are using the paper, there are large subsets of the readership for whom these old papers are standard but we also believe there is good reason readers who are unfamiliar might benefit from the parsimonious ideas in the McRuer model and the like. We hope that our improvements to the modeling description will serve to expose researchers to these useful ideas.

f) The previous study by Kurtzer on long-latency control in cerebellar patients (Kurtzer et al., 2013) was not used in this study since it did highlight a reduction in response gain specific to the long-latency motor system, which is a valuable data for modelling purposes.

We have added the following paragraph to the Discussion which describes the Kurtzer et al. findings and how they relate to the current study:

“Cerebellar patients can also show a reduced feedback gain in situations where responses are driven by proprioception more than vision. Kurtzer et al., 2013, studied cerebellar patients’ ability to modulate long latency stretch responses (i.e. EMG responses within 45-100 ms of a muscle stretch) that depend on knowledge of intersegmental dynamics of the arm. […] These results combined suggest that cerebellar subjects can reweight which feedback modality to rely on (proprioception versus vision) depending on task demands, consistent with Block et al., 2012, and may reduce the sensorimotor gain on either modality to minimize the negative effects of delay.”

3) The results show that the patients lack delay compensation for the predictable visual stimulus. The question arises as whether the problem was delay compensation with respect to their body movements or with respect to the external world. The behavior of Exp 1 is used to conclude that the patients had impairments in their estimate of limb state. This conclusion seems indirect and it is not based on a limb perturbation. The authors imply that delay compensation is associated with proprioceptive estimates only, and that it is not expected to occur in the visual system. The associated rationale is not clear. It would seem that compensation for delays in the visual system is also necessary for controls to track the predictable target. Can we conclude that feedback control is intact, or is this specific form of visuomotor control that is unaffected? Possibly feedback control in the proprioceptive system is also intact, what seems to be missing is delay compensation, but it was not demonstrated whether the faulty delay compensation was in the visual or somatosensory motor system. The lack of predictive control could be in the visual system based on the tracking experiments. The deficit was attributed to predictive control of the limb based on indirect fitting of the model, but the rationale for this is unclear.Related to this issue, the theoretical background suggests a role of the cerebellum in predicting the behavior of one's body (internal model). It is unclear how the experiments test this idea. Experiments 1 and 2 differ in the predictability of the stimulus, not the predictability of the body. The different behavior of the cerebellar patients in experiment 2 (and not in experiment 1) would indicate a problem with the prediction of the world, rather than of the body.

We apologize for the confusion, Experiment 1 was only sum-of-sines (unpredictable) stimuli (see subsection “Generating Target Trajectories (Experiments 1 & 2)”). We perform modeling only on the basis of the sum-of-sines data; that’s why we mainly consider the delay compensation associated with proprioceptive estimates, and do not consider delay compensation for the (unpredictable) target motion in the modeling component of the study. By making comparison between patients and controls in this way, we focused on the role of cerebellum in predicting self-motion from the internal model.

Later we compared between single and sum of sines (see subsection entitled “Experiment 2: Poor Tracking of Simple Oscillatory Trajectories Highlights Cerebellar Patients’ Predictive Deficit” in Results). We find that controls were significantly better than patients at single-sines tracking, illustrating cerebellar patients’ poor predictive ability as compared to controls which seem better able to compensate for visual delay when the target is predictable.

To help avoid confusion about what is tested in what experiments, we have summarized the three type experiments near the beginning of the Materials and methods (subsection “Experimental apparatus, experiments, and task instructions”).

4) It was suggested that cerebellar patients had intact feedback control based on experiment 1. This is likely an overstatement: qualitative differences were visible in the traces of Figure 4 and the interaction between conditions and groups was associated with p = 0.07. Thus, it could be that the relatively low sample size does not allow to conclude for differences based on the analyses suggested.

We agree; we have softened this language. See subsection “Experiment 3: Virtual Reality Feedback Can Reduce Dysmetria”, first paragraph and Discussion, third paragraph. Also note that the Discussion points out that the patient feedback, while intact, is delayed and therefore not fully intact. The point is that the patient population while suffering from imperfect control, much of this seems to be in prediction and delay compensation, but the feedback gains are generally similar.